# IL-33 induces thymic involution-associated naive T cell aging and impairs host control of severe infection

Lei Xu[1,2,3,4,5,6,11], Chuan Wei[1,2,3,4,11], Ying Chen[1,2,3,4], Yue Wu[1,2,3,4], Xiaoli Shou[1,2,3,4], Wenjie Chen[1,2,3,4], Di Lu[1,2,3,4], Haoran Sun[1,2,3,4], Wei Li[6], Beibei Yu[1,2,3,4], Xiaowei Wang[7], Xiaojun Zhang[8], Yanxiong Yu[1,2,3,4], Zhigang Lei[1,2,3,4], Rui Tang[1,2,3,4], Jifeng Zhu[1,2,3,4], Yalin Li[1,2,3,4], Linrong Lu[9], Hong Zhou[10], Sha Zhou[1,2,3,4] ✉, Chuan Su[1,2,3,4] ✉ & Xiaojun Chen[1,2,3,4] ✉

Severe infection commonly results in immunosuppression, which leads to impaired pathogen clearance or increased secondary infection in both humans and animals. However, the exact mechanisms remain poorly understood. Here, we demonstrate that IL-33 results in immunosuppression by inducing thymic involution-associated naive T cell dysfunction with aberrant expression of aging-associated genes and impairs host control of infection in mouse disease models of schistosomiasis or sepsis. Furthermore, we illustrate that IL-33 triggers the excessive generation of medullary thymic epithelial cell (mTEC) IV (thymic tuft cells) in a Pou2f3-dependent manner, as a consequence, disturbs mTEC/cortical TEC (cTEC) compartment and causes thymic involution during severe infection. More importantly, IL-33 deficiency, the anti-IL-33 neutralizing antibody treatment, or IL-33 receptor ST2 deficient thymus transplantation rescues T cell immunity to better control infection in mice. Our findings not only uncover a link between severe infection-induced IL-33 and thymic involution-mediated naive T cell aging, but also suggest that targeting IL-33 or ST2 is a promising strategy to rejuvenate T cell immunity to better control severe infection.

Besides systemic inflammatory response and multiple organ tissue damage, severe infection can also induce immunosuppression[1], which results in impaired pathogen clearance and increased secondary infection[1–3]. For instance, immunosuppression is commonly and closely associated with serious secondary infection in patients with severe sepsis, 2009 H1N1 influenza, COVID-19, schistosomiasis, etc[1,4–7]. Therefore, a more thorough understanding of the mechanisms underlying immunosuppression will facilitate the development of

[1]Jiangsu Key Laboratory of Pathogen Biology, Nanjing Medical University, Nanjing, Jiangsu 211166, P. R. China. [2]State Key Lab of Reproductive Medicine, Nanjing Medical University, Nanjing, Jiangsu 211166, P. R. China. [3]Department of Pathogen Biology and Immunology, Nanjing Medical University, Nanjing, Jiangsu 211166, P. R. China. [4]Center for Global Health, Nanjing Medical University, Nanjing, Jiangsu 211166, P. R. China. [5]Department of Respiratory, Nanjing First Hospital, Nanjing Medical University, Nanjing, Jiangsu 210006, P. R. China. [6]Department of Clinical Laboratory, Nanjing First Hospital, Nanjing Medical University, Nanjing, Jiangsu 210006, P. R. China. [7]Department of Blood Transfusion, Children's Hospital of Nanjing Medical University, Nanjing, Jiangsu 210008, P. R. China. [8]Imaging Center, Children's Hospital of Nanjing Medical University, Nanjing, Jiangsu 210008, P. R. China. [9]Institute of Immunology, School of Medicine, Zhejiang University, Hangzhou 310058, P. R. China. [10]Department of Cell Biology, School of Life Sciences, Anhui Medical University, Hefei 230032, P. R. China. [11]These authors contributed equally: Lei Xu, Chuan Wei. ✉e-mail: shazhou@njmu.edu.cn; chuansu@njmu.edu.cn; chenxiaojun@njmu.edu.cn

better strategies for the clinical intervention of severe infectious diseases.

T cell aging, displaying decreased TCR-triggered proliferation accompanied by restricted TCR repertoire or avidity and enhanced senescence-associated secretory phenotype (SASP), is extensively studied in the aging process, which plays an immunosuppressive role and is responsible for increased susceptibility to infection and decreased vaccination efficacy in older individuals[8]. Interestingly, accumulating evidence has shown that T cell aging-associated dysfunction is accelerated in patients with some infectious diseases, such as HIV and cytomegalovirus infection[9,10]. However, the exact molecular basis of T cell aging-associated dysfunction during severe infection is not fully understood.

In the aging process, one of the primary hallmarks of T cell aging is the involution of the thymus, the major organ responsible for the development, selection, and output of naive T cells[8,11] and the promotion of the lymphopoiesis and immunocompetence of T cells in the peripheral lymphoid organs by providing thymic hormones[12]. In certain stress conditions, such as pregnancy and pathogen infection[13,14], acute thymic involution presents a rapid yet reversible reduction in the size and weight of the thymus, disruption of the thymic epithelial architecture, and depletion of populations of thymocyte subsets[13,15], which is distinct from age-related chronic thymic involution with a gradual progressive decline in cellularity and functions of thymocytes in humans[16]. Indeed, acute thymic involution is a common feature of severe infectious diseases, such as severe schistosomiasis and sepsis[15,17,18]. Although a recent study has suggested the role of acute thymic involution in immune adaptations during pregnancy[14], the exact roles of acute thymic involution during severe infection remain poorly characterized.

Here, by using mouse disease models of schistosomiasis and sepsis, we demonstrated that IL-33-mediated thymic involution resulted in naive T cell dysfunction with aberrant expression of aging-associated genes and impaired host control of severe infection. Further mechanistic studies demonstrated that IL-33 disturbed mTEC/cTEC compartment and consequently caused thymic involution by inducing the excessive generation of mTEC IV (thymic tuft cells) in a Pou2f3-dependent manner. More importantly, IL-33 deficiency, the anti-IL-33 neutralizing antibody treatment, or ST2 deficient thymus transplantation restored T cell immune responses and more effectively controlled infection. Our findings not only demonstrate a mechanism underlying naive T cell dysfunction associated with aging during severe infection but also suggest that IL-33 or ST2 is a promising intervention target to rejuvenate T cell function to better control severe infection.

## Results

### Schistosome infection induces naive CD4+ T cell aging in mice
To dissect the effect of severe infection on host naive CD4+ T cells, we employed schistosome infection, a helminthiasis affecting approximately 200 million people[19]. Naive CD4+ T cells were isolated by magnetic-activated cell sorting (MACS) from the spleen of uninfected mice or schistosome-infected mice and were performed RNA-sequencing (RNA-seq) experiments. RNA-seq analysis revealed that severe schistosome infection upregulated 3,450 genes in naive CD4+ T cells (fold change ≥ 2, q ≤ 0 .05), while it repressed 2,567 genes (Fig. 1a). Notably, naive CD4+ T cells from schistosome-infected mice displayed decreased expression of the double-strand-break repair nuclease MRE11A (encoded by *Mre11a*) and increased expression of the DNA damage marker γH2aX (encoded by H2ax) (Fig. 1b), suggesting increased DNA damage in these cells[8]. Naive CD4+ T cells from schistosome-infected mice upregulated the expression of the senescence markers cyclin-dependent kinase inhibitor 1

(CDKN1A/p21, encoded by *Cdkn1a*) and cyclin-dependent kinase inhibitor 2 A (CDKN2A/P16, encoded by *Cdkn2a*), downregulated the expression of Cyclin D (encoded by *Ccnd1*) and Cyclin E (encoded by *Ccnd2*), and lost the expression of the costimulatory molecules CD27 (encoded by *Cd27*) and CD28 (encoded by *Cd28*) (Fig. 1b), supporting that severe schistosome infection induces a senescent-like phenotype in naive CD4+ T cells[8]. Strikingly, the expression of the mitochondrial transcription factor A (TFAM, encoded by *Tfam*) was reduced in naive CD4+ T cells from schistosome-infected mice (Fig. 1b), which may cause an immunometabolic dysfunction, and as a consequence, drive T cell aging[20]. Consistently, gene set enrichment analysis (GSEA) of differentially expressed genes showed significant enrichment of the glycolysis pathway-associated gene signature in naive CD4+ T cells from schistosome-infected mice (Fig. 1c), accompanied by an enhanced senescence-associated secretory phenotype (SASP), as shown by increased expressions of *Ccl1*, *Ccl2*, *Ccl3*, *Il6*, *Tnf*, *Csf1*, *Gzmb*, and *Gzmk* (Fig. 1d, e). Meantime, the expression of Foxo1 was substantially reduced in naive CD4+ T cells from schistosome-infected mice (Fig. 1f, g), suggesting a T cell aging-associated loss of proteostasis[8]. More importantly, the expression levels of CD5 on naive T cells in the spleen and blood, which have been widely used to reflect the strength and/or duration of TCR signaling in naive CD4+ T cells and CD8+ T cells in the thymus and periphery[21-24], were dramatically decreased in schistosome-infected mice (Fig. 1h–j and Fig. S1), indicating that these naive T cells may poorly respond to antigens, such as schistosome specific antigens and bystander antigens. Supportively, results showed that peripheral naive CD4+ T cells from schistosome-infected mice displayed impaired proliferation when polyclonally activated with an anti-CD3 mAb (Fig. 1k, l), which demonstrates that severe schistosome infection induces naive CD4+ T cell aging in mice.

### Naive T cell aging is linked to thymic involution during severe schistosome infection
Given that thymic involution is a primary hallmark of T cell aging[8] and has been observed in schistosome-infected mice[17], we next investigated whether thymic involution contributed to naive T cell aging-associated dysfunction during severe infection. Results showed that the size and weight of the thymus (Fig. 2a, b) and the number of thymocytes (Fig. 2c) were remarkably reduced in mice starting at week 5–6 after infection, accompanied by an aberrant development of T cells (Fig. 2d), reduced expression of CD5 (Fig. S2a–c), and decreased proliferation of naive T cells (Fig. 2e, f). In addition, the apoptosis of thymocytes including CD4+CD8+ double-positive (DP), CD4+ single-positive (CD4SP), CD8+ single-positive (CD8SP), and CD4-CD8- double-negative (DN) thymocytes, was strongly increased in schistosome-infected mice (Fig. S2d, e). In contrast, thymic involution was abrogated in mice 7 weeks after antischistosomal treatment, as shown by the fact that antischistosome-treated mice displayed normal thymic size, weight, cellularity, and thymocyte subsets (Fig. 2g–j). In parallel with abolished thymic involution, the expression of CD5 on naive T cells was restored in infected mice 7 weeks after antischistosomal treatment almost up to normal levels (Fig. S2f–h), which was concomitant with increased naive T cell proliferation (Fig. 2k, l). Taken together, these results suggest that T cell aging-associated dysfunction is linked to thymic involution during severe schistosome infection.

### IL-33 results in thymic involution during severe infection
Substantial evidence shows that the alarmin IL-33 was strongly produced by inflammation-damaged tissues in severe infectious diseases

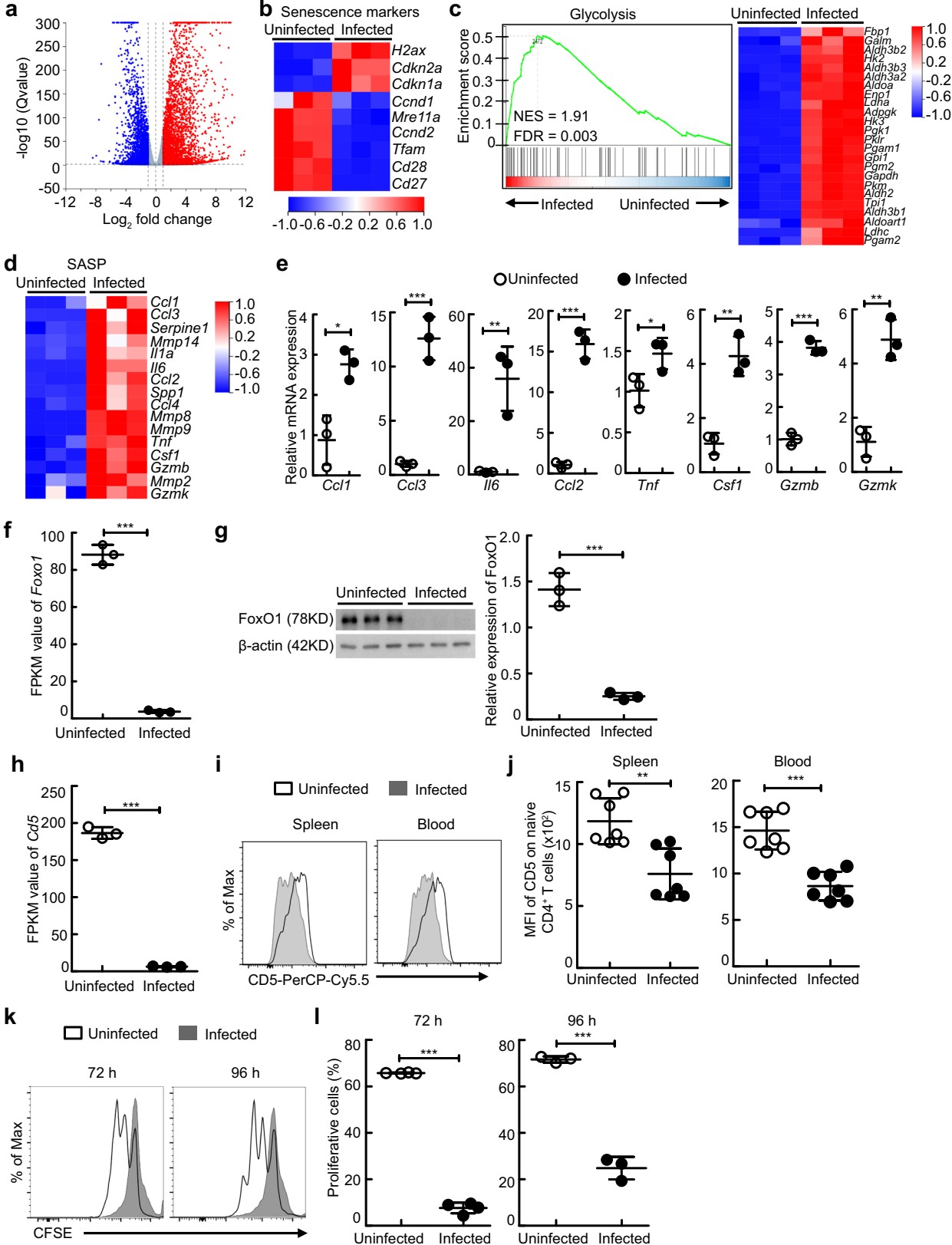

such as schistosomiasis and sepsis[25–27]. As expected, our results showed that the level of full-length or short-form of IL-33 was markedly elevated in the liver in schistosome-infected mice (Fig. S3a), where schistosome eggs provoked a vigorous granulomatous inflammatory response. In contrast, we observed that the level of the full-length of IL-33 but not the short-form of IL-33 was substantially increased in the thymus of mice with schistosomiasis (Fig. 3a), which may be due to a low level of cleaved IL-33 or lack of cleaving proteases of IL-33 in the thymus during schistosome infection[28]. Next, we wondered whether the elevation of IL-33 was involved in thymic involution during severe

**Fig. 1 | Schistosome infection induces T cell aging in mice. a–d, f, j** Naive CD4[+] T cells were sorted from uninfected or schistosome-infected mice ($n = 3$ mice) to perform RNA-seq analysis. **a** Volcano plot shows genes differentially expressed in naive CD4[+] T cells. **b** Heat map shows the genes of senescence markers. **c** Gene set enrichment analysis (GSEA) of glycolysis pathway-associated genes in naive CD4[+] T cells (left) and heat map of selected genes (right). **d** Heat map shows the genes of senescence-associated secretory phenotype (SASP). **e** *Ccl1, Ccl3, Il6, Ccl2, Tnf, Csf1, Gzmb,* and *Gzmk* gene expressions of naive CD4[+] T cells were determined by RT-PCR ($n = 3$ mice). *Ccl1, P = 0.0104; Ccl3, P = 0.0006; Il6, P = 0.0075; Ccl2, P = 0.0002; Tnf, P = 0.0481; Csf1, P = 0.0025; Gzmb, P < 0.0001; Gzmk, P = 0.0021.* **f** FPKM value of *Foxo1* gene of naive CD4[+] T cells ($n = 3$ mice); $P < 0.0001$. **g** Representative and quantified western blots of FoxO1 in naive CD4[+] T cells ($n = 3$ mice); $P < 0.0001$. **h** FPKM value of *Cd5* gene of naive CD4[+] T cells ($n = 3$ mice); $P < 0.0001$. **i–l** Cells were from uninfected mice or mice 8 weeks after schistosome infection. **i, j** Representative and quantified flow cytometry of the mean fluorescence intensity (MFI) of CD5 on naive CD4[+] T cells from the spleen or peripheral blood ($n = 7$ mice, pool of two independent experiments); Spleen, $P = 0.0016$; Blood, $P < 0.0001$. **k, l** Representative and quantified flow cytometry of the CFSE MFI of CFSE-labeled naive CD4[+] T cells from spleen stimulated with anti-CD3 and anti-CD28 antibodies for 72 or 96 h ($n = 4$ mice for 72 h; $n = 3$ mice for 96 h); 72 h or 96 h, $P < 0.0001$. All data are shown as the mean ± s.d. *$P < 0.05$, **$P < 0.01$, ***$P < 0.001$, Unpaired two-tailed Student's *t*-test. Source data are provided as a Source Data file.

schistosome infection. Concomitant with reversed thymic phenotype and function in mice 7 weeks after antischistosomal treatment shown in the aforementioned data (Fig. 2), antischistosomal treatment reduced accumulation of IL-33 in the thymus from schistosome-infected mice (Fig. 3a). More importantly, IL-33 deficient mice displayed normal thymic morphology and cellularity (Fig. 3b–d) and thymocyte development (Fig. 3e) after schistosome infection, which was coincident with decreased thymocyte apoptosis (Fig. S3b). IL-33 neutralization (Fig. 3f–i) or ST2 deficiency (Fig. S3c–e) phenocopied IL-33 deficiency, indicating a major role of extracellular IL-33, rather than nuclear IL-33, in thymic involution through the membrane-bound receptor ST2 during schistosome infection. Interestingly, the serum level of sST2 was significantly increased in mice after schistosome infection (Fig. S3f), indicating a potential regulatory role of sST2 in IL-33-mediated thymic involution. In addition, sepsis, a severe systemic infectious disease, also resulted in increased IL-33 level in the thymus (Fig. S3g), and as a consequence, caused thymic involution in mice and humans (Fig. S3h–k), while IL-33 deficient mice displayed the normal morphology, weight, and cellularity of the thymus during sepsis (Fig. S3h–j).

To next determine if IL-33 alone was sufficient to induce thymic involution, we injected recombinant IL-33 or vehicle (PBS) intraperitoneally into normal WT mice. Results showed that administration of IL-33 led to distinct reductions in the size, weight, and cellularity of the thymus (Fig. 3j–l) and abnormal thymocyte development (Fig. 3m), which was coincident with increased thymocyte apoptosis (Fig. S3l), while ST2 deficiency in mice abolished the effect of IL-33 administration on thymic involution (Fig. S3m–o).

To further determine whether IL-33-mediated thymic involution was thymus intrinsic in vivo, we established thymus transplantation under the kidney capsule (Fig. S3p–r). Results showed that the IL-33 receptor (ST2)-sufficient recipient or donor thymus had undergone profound involution, while the ST2-deficient donor thymus was normal in the morphology, weight, and cellularity in IL-33-treated or schistosome-infected recipient mice (Fig. 3n–q). Taken together, these results demonstrate that IL-33 leads to thymic involution in a thymus-intrinsic manner during severe infection.

## IL-33-mediated thymic involution induces naive T cell aging and impairs host control of infection

Next, we explored the consequences of IL-33-induced thymic involution on T cell immunity during severe infection. Concomitant with restored thymic phenotype and function, the TCR avidity of naive T cells was markedly higher in *Il-33[-/-]* mice with schistosomiasis or sepsis than that in wildtype mice, as shown by a reduced decline in expression levels of CD5 on naive T cells in thymus, spleen, and blood in *Il-33[-/-]* mice (Fig. S4a–c) or anti-IL-33-treated mice (Fig. S4d–f) with schistosomiasis, or *Il-33[-/-]* mice with sepsis (Fig. S4g–i). Accordingly, naive CD4[+] T cells from *Il-33[-/-]* or anti-IL-33 antibody-treated infected mice exhibited an increased proliferation than those from wildtype infected mice (Fig. 4a–d).

In contrast, administration of IL-33 resulted in naive-memory T cell imbalance (Fig. S5a–d), a decrease in expression levels of CD5 on naive T cells in the thymus, spleen, and blood (Fig. S5e–g), and attenuated responsiveness of peripheral naive CD4[+] T cells to the anti-CD3 mAb (Fig. 4e, f).

To further investigate whether IL-33 impaired T cell immunity by inducing thymic involution, ST2 sufficient or deficient thymus was transplanted into wildtype mice treated with IL-33 or infected with schistosome. The results showed that the transplantation of IL-33 receptor (ST2)-deficient thymus strongly restored T-cell proliferative activity from mice treated with IL-33 or infected with schistosome (Fig. 4h, i, k, and l), concomitant with increased expression levels of CD5 on naive T cells in the thymus, spleen, and blood (Fig. 4g, 4j, and Fig. S5h–k). More importantly, the transplantation of ST2-deficient thymus, but not ST2-sufficient thymus, significantly promoted T cell-mediated egg granuloma formation to protect host tissues from egg-derived toxins in recipient mice infected with schistosome (Fig. 4m, n)[29], as well as lengthened the survival of recipient mice with sepsis (Fig. 4o). Taken together, these results demonstrate that IL-33-mediated thymic involution impairs host T-cell immune response against pathogens during severe infection, which further suggests that IL-33 and/or its receptor ST2 could be a promising target for intervention to reverse thymic involution and restore T-cell immunity.

## IL-33 perturbs the compartment of thymic epithelial cells both in vitro and in mice with IL-33 administration or severe infection

Given that the abnormality in function or compartment of TECs is a leading cause of thymic involution[30], we next wondered whether IL-33 had an impact on TEC function or compartment. To dissect the effect of IL-33 on TECs in vitro, we isolated TECs by MACS from thymus cultured in fetal thymic organ cultures (FTOC) treated with IL-33 or PBS (Fig. S6a), and performed RNA-sequencing (RNA-seq) experiments. RNA-seq analysis revealed that IL-33 upregulated 457 genes in TECs (fold change ≥ 2, q ≤ 0.05), while it repressed 1,702 genes (Fig. 5a). Strikingly, IL-33 altered the expression of TEC function-related genes (Fig. 5b), such as *Foxn1, Cd83,* and *prdm1* (required for mTEC or cTEC functions), *Aire* and *Fezf2* (required for tissue-restricted antigen generation), *Ccl19, Ccl25, Xcl1, Ccl22,* and other chemokines (required for the trafficking of thymic progenitors, thymocytes, or thymic DCs), *Ctsh* (required for negative selection of T cells), and *Tmsb15b2* and *Tmsb15a* (encoding thymosin β15a and thymosin β15b2, respectively; important regulators of the peripheral immune response), which suggested that IL-33 altered the functions of TECs.

Because of the central role of mTEC/cTEC compartments in thymic involution[30], we next determined whether IL-33 disturbed mTEC/cTEC compartments during IL-33-mediated thymic involution. As expected, the mTEC-specific genes, such as *Ascl1, Ehf, Ccl19,*

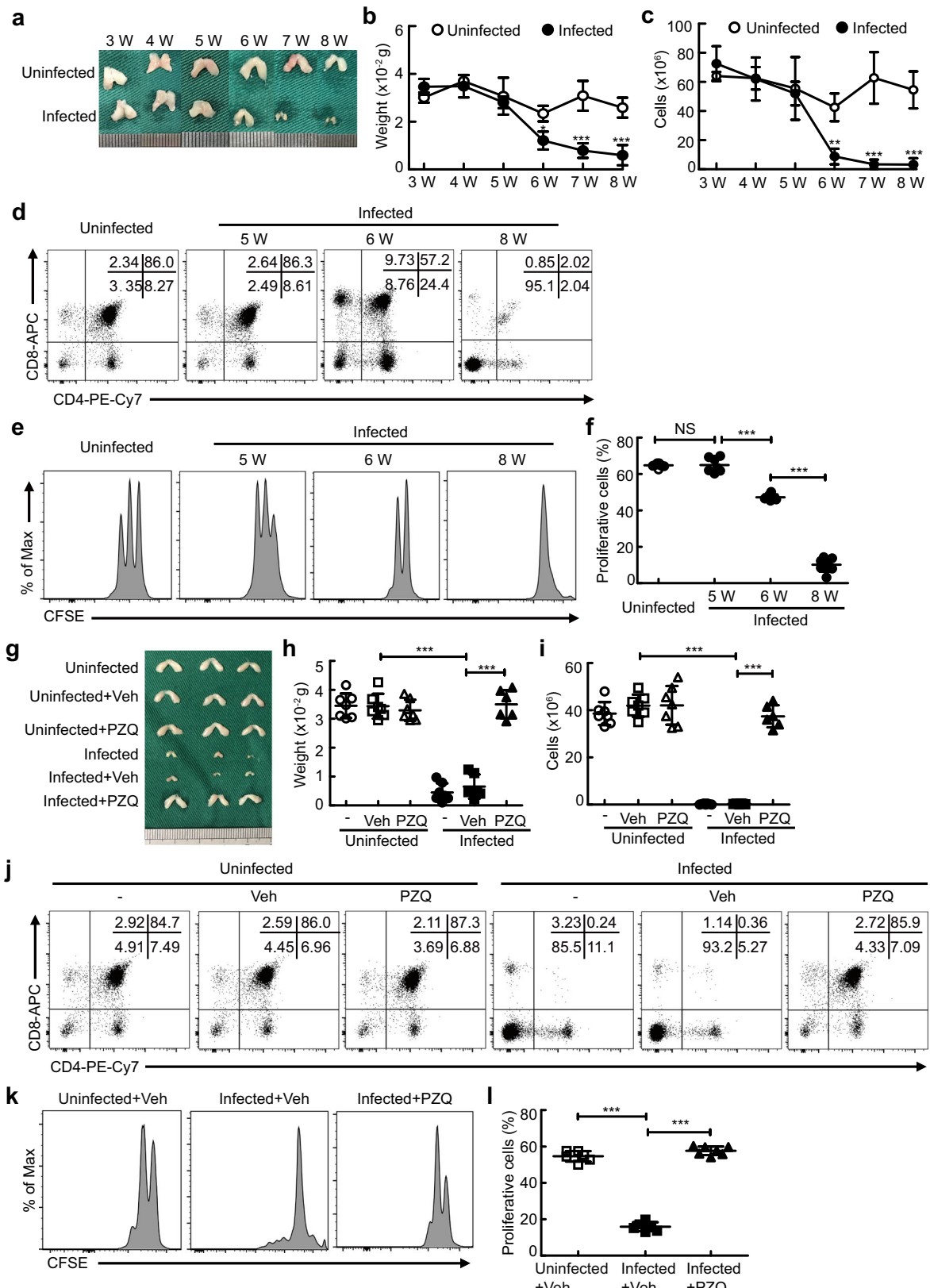

*Pigr, Trpm5, Avil, Gnat3, Plcb2, Chat, Alox5, Ltc4s, Pde2a, Gnb3*, and *Lrmp*, were upregulated in IL-33-treated TECs, while cTEC-specific genes, such as *Enpep* (encoding LY51), *Ccl25, Ackr4* (encoding Ccrl1), *Cd83, Foxn1*, and *Tbeta*, were downregulated (Fig. 5c). Of note, *Enpep* (encoding LY51), a specific marker of cTECs[31], was

dramatically decreased in IL-33-treated TECs (Fig. 5d), which indicated that IL-33 disrupted mTEC/cTEC compartments. Consistent with RNA-seq analysis, flow cytometry analysis showed an increase in mTEC population, but a decrease in cTEC population in the thymus treated with IL-33 in vitro in FTOC (Figs. 5e–g), while

**Fig. 2 | Schistosome infection-induced thymic involution links to T cell aging in mice. a–c** Representative morphology and quantified weight and cellularity of thymus from uninfected mice or mice 3, 4, 5, 6, 7, or 8 weeks after infection ($n = 4$ mice). 6 W, $P = 0.0265$ (weight) or 0.0091 (cells); 7 W, $P < 0.0001$ (weight) or <0.0001 (cells); 8 W, $P < 0.0001$ (weight) or <0.0001 (cells); Two-way ANOVA with Tukey's multiple comparisons. **d** Representative flow cytometry of thymocytes. **e, f** Representative and quantified flow cytometry of the CFSE MFI of naive CD4$^+$ T cells stimulated with anti-CD3 and anti-CD28 antibodies (5 W, $n = 7$ mice; other groups, $n = 8$ mice; pool of two independent experiments). 5 W versus uninfected, $P = 0.9995$; 6 W versus 5 W, $P < 0.0001$; 8 W versus 6 W, $P < 0.0001$; One-way ANOVA with Tukey's multiple comparisons. **g–i** Mice were treated with PZQ at week 8 after infection and sacrificed 7 weeks after PZQ treatment. **g–i** Representative morphology and quantified weight and cellularity of thymus (Infected+Veh or PZQ,

$n = 6$ mice; Uninfected, Uninfected+Veh, Uninfected+PZQ, or Infected, $n = 7$ mice; pool of two independent experiments). Infected+Veh versus uninfected+Veh, $P < 0.0001$ (weight), $P < 0.0001$ (cells); Infected+PZQ versus Infected+Veh, $P < 0.0001$ (weight), $P < 0.0001$ (cells); One-way ANOVA with Tukey's multiple comparisons. **j** Representative flow cytometry of thymocytes. **k, l** Representative and quantified flow cytometry of the CFSE MFI of CFSE-labeled naive CD4$^+$ T cells stimulated with anti-CD3 and anti-CD28 antibodies (Uninfected+Veh or Infected +Veh, $n = 6$ mice; Infected+PZQ, $n = 7$ mice; pool of two independent experiments). Infected+Veh versus uninfected+Veh, $P < 0.0001$; Infected+PZQ versus Infected +Veh, $P < 0.0001$; One-way ANOVA with Tukey's multiple comparisons. All data are shown as the mean ± s.d; *$P < 0.05$, **$P < 0.01$, ***$P < 0.001$, NS, not significant. Source data are provided as a Source Data file.

ST2 deficiency abolished IL-33-mediated mTEC/cTEC imbalance (Fig. S6b–d). Furthermore, IL-33 administration also perturbed mTEC/cTEC compartments in vivo (Fig. 5h–k), while ST2 deficiency abolished IL-33-mediated mTEC/cTEC imbalance (Fig. S6e–h). More importantly, schistosome infection or sepsis also resulted in abnormal mTEC/cTEC compartments, as shown by an increased proportion of mTECs but a decreased proportion of cTECs, while IL-33 or ST2 deficiency, or anti-IL-33 treatment restored mTEC/cTEC compartments in mice with schistosomiasis (Fig. 5l–r, S6i–l) or sepsis (Fig. S6m–o). Taken together, these results show that IL-33 disrupts the compartment of thymic epithelial cells, which is a leading cause of thymic involution.

### IL-33 disturbs the TEC compartment by inducing excessive mTEC differentiation in a thymocyte-independent manner

To study how IL-33 played a pathogenic role in disturbing the TEC compartment, we examined the expression of ST2 on cells in the thymus. Flow cytometry analysis showed that a higher percentage of TECs expressed ST2 compared with thymocytes, non-TEC stromal cells, or Treg cells (Fig. 6a, b, and Fig. S7a–e), whereas the MFI of ST2 was comparable on these subsets of cells in the thymus during schistosome infection (Fig. S7f). Indeed, no direct effect of IL-33 on thymocyte apoptosis was observed in vitro (Fig. 6c). To further rule out the possibility that thymocytes were involved in IL-33-induced mTEC accumulation, deoxyguanosine (dGUO) was employed to eliminate thymocytes in the thymus. Of note, IL-33 resulted in an excessive increase in the mTEC population in the absence of thymocytes in the dGUO-treated thymus in FTOCs (Fig. 6d–f). These results, in conjunction with our observation that donor dGUO-treated WT thymus, rather than donor dGUO-treated ST2 deficient thymus, displayed severe involution in thymus-transplanted mice treated with IL-33 or infected with schistosome (Fig. 3n–q), demonstrate that IL-33 results in aberrant accumulation of mTECs and subsequent thymic involution independently of thymocytes.

Next, we further wondered how IL-33 led to an excessive increase in the ratio of mTEC/cTEC. Intriguingly, there was no noticeable difference in IL-33-induced apoptosis or proliferation between mTECs and cTECs (Fig. 6g, h), raising the possibility that IL-33-mediated mTEC accumulation may be through affecting the differentiation of TEC into mTECs. As expected, the level of the noncanonical NF-κB transcription factor p100/p52, which plays a key role in mTEC differentiation[32], was elevated in TECs treated with IL-33 (Fig. 6i). Inhibition of NF-κB-inducing kinase (NIK), a central component of the noncanonical NF-κB pathway, completely abolished the IL-33-mediated mTEC accumulation (Fig. 6j–l). Taken together, these results suggest that IL-33 disturbs the TEC compartment by inducing aberrant differentiation of TECs into mTECs to alter the proper ratio of mTEC/cTEC, which is a leading cause of thymic involution.

### IL-33-induced generation of mTEC IV contributes to an aberrant increase in the mTEC population and subsequent thymic involution

Given the heterogeneity of mTECs, consisting of four major mTEC subpopulations (mTEC I–IV) with distinct transcriptional profiles and lineage regulators[33,34], we next wondered whether IL-33 promoted excessive accumulation of the mTEC population by inducing the aberrant generation of a particular subset of mTECs. RNA-seq analysis showed that in TECs from the thymus in FTOCs treated with IL-33 for 4 days, mTEC IV (thymic tuft cell)-specific genes, such as *Trpm5*, *Pou2f3*, *Avil*, *Gnat3*, *Plcb2*, *Chat*, *Alox5*, *Ltc4s*, *Pde2a*, *Gnb3*, and *Lrmp*, were globally upregulated (Fig. 7a). Meanwhile, mTEC II-specific genes including *Ctsh*, *Aire*, *Fezf2*, *Klk1b16*, *Apoc2*, *Cyp24a1*, and *Muc3*, and mTEC III-specific genes including *Fgf21*, *Krt17*, *Spink5*, *Plb1*, *Cd80*, *Cd86*, *Krt1*, *Krt77*, and *Asprv1*, were globally downregulated (Fig. 7a). In contrast, mTEC-I specific genes, such as *Six4*, *Chuk*, *Bmp4*, *Itga6*, *Itgb4*, *Krt5*, *Sox4*, and *Ly6a*, displayed no significant change (<2 fold change) (Fig. 7a).

To further confirm that IL-33 promoted excessive accumulation of TECs by inducing the aberrant generation of mTEC IV cells (thymic tuft cells), we detected the expression level of the transcription factor Pou2f3 in TECs, a master regulator of mTEC IV cells[33,34], and found a dramatic increase in Pou2f3 expression in TECs treated with IL-33 (Fig. 7b, c). Consistently, flow cytometric analysis showed that mTEC IV cells were indeed induced preferentially in the thymus treated with IL-33 in FTOCs (Fig. 7d, e). Strikingly, the percentage of mTEC I cells was also increased after the accumulation of mTEC IV cells in the thymus in FTOCs with IL-33 treatment for 7 days, while the percentages of mTEC II cells and mTEC III cells were decreased (Fig. 7e). Consistent with these observations in vitro, the percentages of mTEC IV cells and mTEC I cells were also increased in mice injected with IL-33 or infected with schistosome, while the percentages of mTEC II cells and mTEC III cells were decreased (Fig. 7f, g). In contrast, thymic deletion of *pou2f3* resulted in reduced accumulation of mTECs by IL-33 treatment (Fig. 7h–j). Meanwhile, Pou2f3 deficient mice displayed reduced accumulation of mTECs (Fig. 7k–m). More importantly, Pou2f3 deficiency alleviated thymic involution (Fig. 8a–c), restored naive T cell proliferation (Fig. 8d, e), and promoted T cell-mediated granuloma formation (Fig. 8f, g) in mice after schistosome infection. Taken together, these results indicate that IL-33-induced excessive generation of mTEC IV cells (thymic tuft cells) contributes to the disruption of TEC compartment by changing the ratio of mTEC/cTEC and subsequent thymus involution-mediated T cell aging during severe infection.

### Discussion

The immune system in a host with severe infection may lose its ability to mount an effective immune response to destroy or constrain pathogens due to T cell aging-associated immunosuppression.

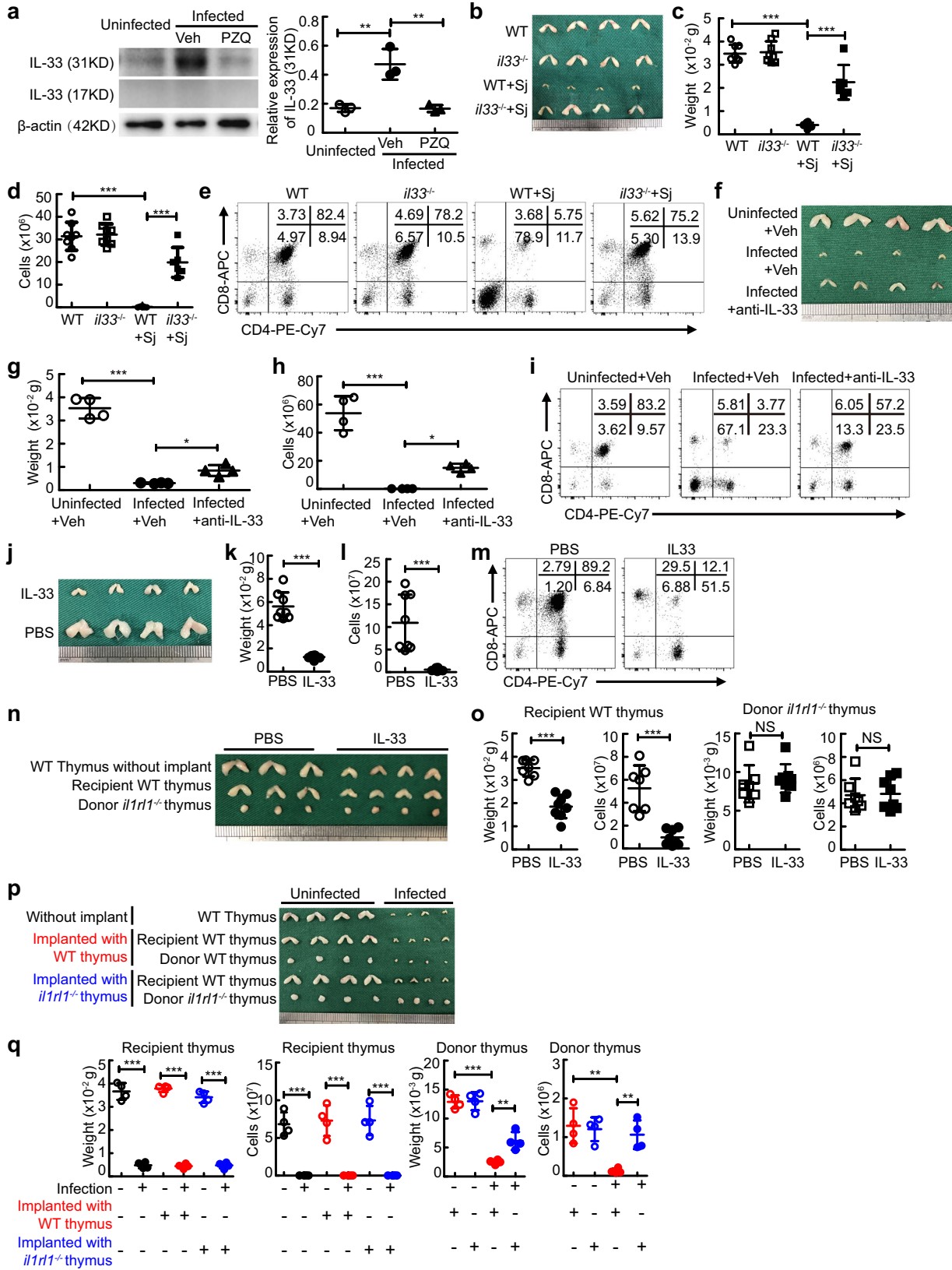

Although T cell aging is a common feature of severe infection, the mechanisms underlying this process are not fully understood. Our study reveals that infection-induced IL-33 production causes T cell aging by inducing thymic involution and consequently impairs host control of severe infection. Furthermore, IL-33 leads to host thymic involution by disrupting mTEC/cTEC compartments, which is due to excessive generation of mTEC IV (thymic tuft cells) and subsequent mTEC I cells.

Either T cell immune deficiency or increased susceptibility to infections has been reported to relate to impaired thymus

**Fig. 3 | IL-33 causes thymic involution during schistosome infection.**
**a** Representative and quantified western blots of IL-33 (*n* = 3 mice). Infected+Veh versus uninfected or Infected+PZQ, *P* = 0.0029 or *P* = 0.0027. **b**–**d** Representative morphology and quantified weight and cellularity of the thymus (*il33*−/−+Sj, *n* = 6 mice; other groups, *n* = 8 mice; pool of two independent experiments), *P* < 0.0001. **e** Representative flow cytometry of thymocytes. **f**–**i** Mice were treated with anti-IL-33 at week 3 every other day until week 8 after infection. **f**–**h** Representative morphology and quantified weight and cellularity of thymus (*n* = 4 mice). Infected +Veh versus Uninfected+Veh or Infected+anti-IL-33, *P* < 0.0001 (weight or cells) or *P* = 0.0461 (weight), *P* = 0.0411 (cells). **i** Representative flow cytometry of thymocytes. **j**–**m** Mice were injected with IL-33 for six consecutive days. **j**–**l** Representative morphology and quantified weight and cellularity of the thymus (*n* = 8 mice, pool of two independent experiments). *P* < 0.0001 (weight), *P* = 0.0003 (cells). **m** Representative flow cytometry of thymocytes. **n**, **o** Representative morphology

(**n**) and quantified weight and cellularity (**o**) of thymus (PBS, *n* = 7 mice; IL-33, *n* = 8 mice; pool of two independent experiments). Recipient WT thymus, *P* < 0.0001 (weight or cells); Donor *il1rlI*−/− thymus, *P* = 0.5391 (weight), *P* = 0.8775 (cells). **p**, **q** Representative morphology (**p**) and quantified weight and cellularity (**q**) of thymus (*n* = 4 mice). Infected versus Uninfected, *P* < 0.0001 (weight or cells) for recipient thymus; Infected+WT thymus versus Uninfected+ WT thymus, *P* < 0.0001 (weight or cells) for recipient thymus, *P* < 0.0001 (weight) or *P* = 0.0016 (cells) for donor thymus; Infected+*il1rlI*−/− thymus versus Uninfected+*il1rlI*−/− thymus, *P* < 0.0001 (weight or cells) for recipient thymus, *P* = 0.0071 (weight) or *P* = 0.0085 (cells) for donor thymus. All data are shown as the mean ± s.d; *P* < 0.05, **P* < 0.01, ***P* < 0.001, NS, not significant. **a**, **c**, **d**, **g**, **h**, **q** One-way ANOVA with Tukey's multiple comparisons; **k**, **l**, **o** Unpaired two-tailed Student's *t*-test. Source data are provided as a Source Data file.

function in children with DiGeorge syndrome, severe combined immunodeficiency, or in preterm newborn children[35]. In addition, thymectomized adult subjects also display deficient T-cell repertoires with limited capacity to respond to new antigens[36,37]. On the contrary, myasthenia gravis in adult patients improve clinically after thymectomy[37]. These findings support the notion that the thymus is important for optimal T cell immunity throughout life, even very late in life[37,38].

Chronic thymic involution during aging has plausible biological functions from optimizing peripheral repertoire to conserving energy, or causing T-cell immunological aging[39–41]. Although one recent report reveals a physiological role for acute thymic involution in immune adaptations in pregnancy to safeguard fetal development[14], the biological functions of acute thymic involution during severe infection remain largely obscure. Here, we reveal a pathological role for acute thymic involution in severe infection-induced T cell aging in mouse models of schistosomiasis and sepsis. More importantly, our retrospective results showed that sepsis patients display severe thymic involution. Our findings, in conjunction with the previous reports that the activation, proliferation, and thymic output of naive T cells in sepsis patients are impaired[42,43], suggest that acute thymic involution may also contribute to compromised T-cell immunity and impaired control of infection in patients with severe infectious diseases. These data indicate that reversing thymic involution is a potential therapeutic strategy for rejuvenating T cell immune responses to better control severe infection.

IL-33 functions as a nuclear transcriptional regulator or a cytokine released into the extracellular space after cell injury[44,45]. In this report, we demonstrate that infection-triggered IL-33 release results in thymic involution-mediated T cell aging in an ST2-dependent manner, suggesting that IL-33 functions as a cytokine, rather than an intracellular nuclear transcriptional regulator, to cause thymic involution. IL-33 can be expressed in many types of cells, such as epithelial cells, endothelial cells, fibroblasts, macrophages, natural killer T cells, and regulatory T cells[28]. IL-33 lacks a secretion signal and thus does not follow the conventional route of secretion via the endoplasmic reticulum-Golgi apparatus secretory pathway[28]. Besides cellular damage, accumulating evidence support that membrane pores are also involved in the release of IL-33 from cells[28]. For instance, recent reports show that IL-33 can be released from dendritic cells, intestinal epithelial cells, and hepatic stellate cells via membrane pores driven by perforin-2, gasdermin C, and gasdermin D, respectively[46–48]. However, the major cellular source and exact release mechanism of IL-33 during infection-induced thymic involution warrant further investigation. Although future studies are still needed to ascertain the relationships between IL-33 and other contributors to thymic involution during severe infection, such as TNF receptor superfamily-related signals and pathogen-associated molecular patterns[14,49], our results indicate that clinical intervention targeting IL-33 or ST2 by anti-IL-33 or anti-ST2 antibody drugs, such as

MEDI3506 (anti-IL-33), Etokimab (anti-IL-33, ANB020), or Astegolimab (anti-ST2, MSTT1041A)[50,51], maybe a promising therapeutic strategy for reversing thymic involution and T cell function to better control severe infection. Indeed, a recent randomized controlled trial shows that MEDI3506 (anti-IL-33) has led to a marked reduction in severe respiratory failure and better control of infection in patients with COVID-19[52]. Whether MEDI3506-mediated better control of infection in these patients is (at least partially) associated with the reverse of thymic involution-mediated T cell aging remains to be further investigated.

It's well known that the disruption of the compartments of mTECs/cTECs is a leading cause of thymic involution[30]. Our data demonstrate that IL-33 results in an aberrant increase in the mTEC population independently of thymocytes, disrupting the mTEC/cTEC compartments. We cannot exclude the possibility that non-TEC stromal cells, such as thymic fibroblasts and endothelial cells[53,54], maybe also involved in IL-33-mediated excessive generation of mTECs and acute thymic involution. However, given that TECs rather than non-TEC stromal cells express a high level of ST2 and that several studies have revealed the direct roles of IL-33 in other epithelial cells, such as airway epithelial cells, esophageal epithelial cells, and intestinal epithelial cells[55,56], it is thus plausible that IL-33 may directly result in aberrant differentiation of TECs into mTECs.

Recent studies have shown that mTECs are highly heterogeneous and comprise four major subpopulations, including mTEC I cells, mTEC II cells, mTEC III cells, and mTEC IV cells (thymic tuft cells), a new tuft-like mTEC subpopulation controlled by the transcription factor Pou2f3[11,33,34]. Here, we have demonstrated that IL-33 disrupts mTEC/cTEC compartments by inducing the excessive generation of mTEC IV and mTEC I cells. Strikingly, IL-33 preferentially promotes mTEC IV cell generation in a Pou2f3-dependent manner before affecting the accumulation in mTEC I cells. Nevertheless, the exact mechanism of how IL-33-mediated excessive generation of mTEC IV cells causes the accumulation in mTEC I cells remains to be addressed.

In summary, we demonstrate that IL-33 results in thymic involution-mediated naive T cell aging and impairs host control of severe infection. Our further mechanistic studies show that IL-33 disturbs mTEC/cTEC compartment and consequently causes thymic involution by inducing the excessive generation of mTEC IV in a Pou2f3-dependent manner. Our findings suggest that targeting IL-33 or ST2 may be a promising intervention avenue for rejuvenating T-cell immunity to better control severe infection.

## Methods
### Study approval
All animal procedures were conducted following the Regulations for the Administration of Affairs Concerning Experimental Animals (1988.11.14) and approved by the Institutional Animal Care and Use

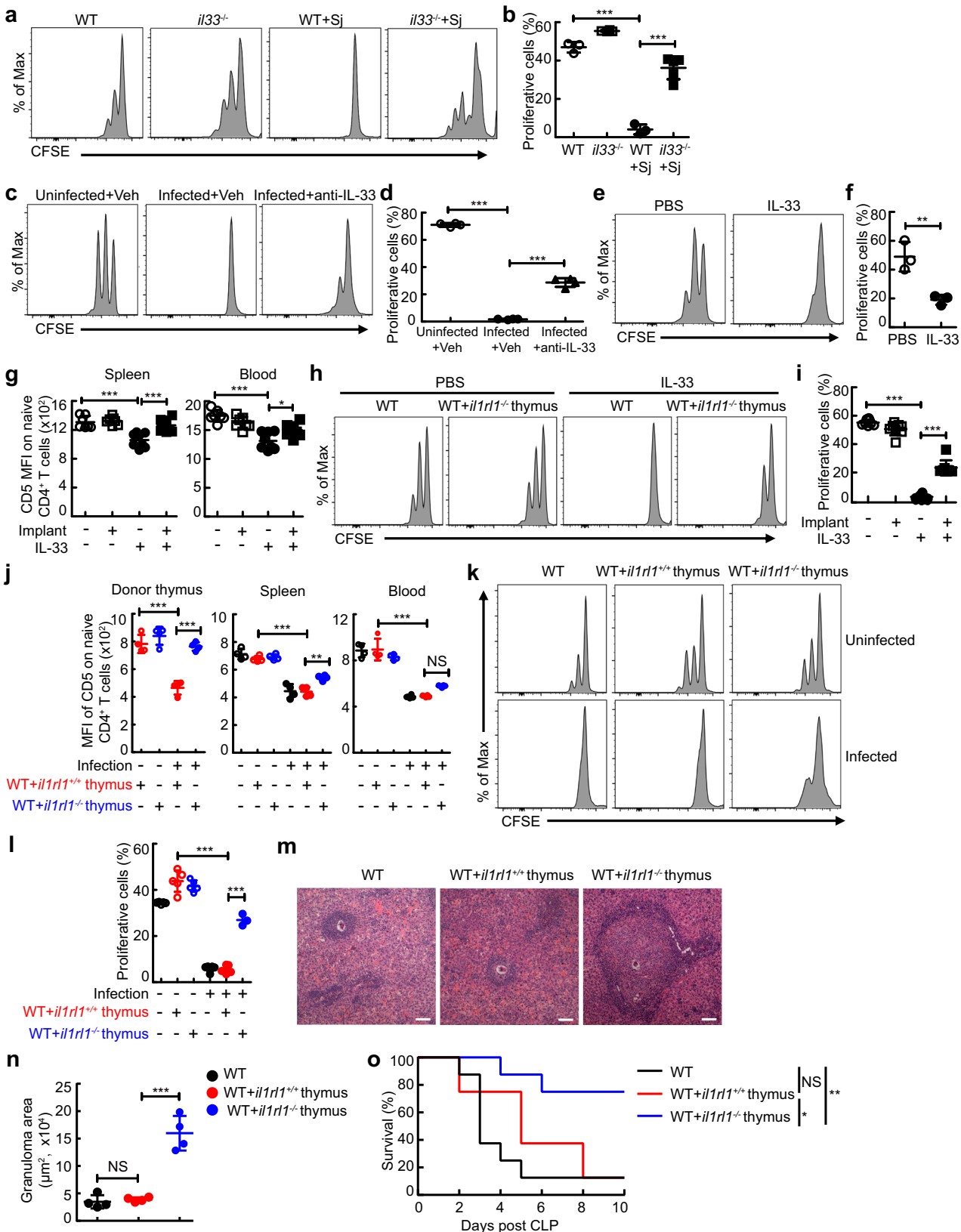

Committee (IACUC) for the use of laboratory animals at Nanjing Medical University (Permit Number: IACUC-1712022). Ethical clearance for this retrospective study was obtained from the Institutional Review Board of Nanjing Medical University, Nanjing, China (Permit Number: 2020607). Informed consent was obtained after the nature, and possible consequences of the studies were explained.

**Mice**

Six to eight-week-old wild-type C57BL/6 J mice and nude mice were purchased from the Animal Core Facility of Nanjing Medical University (Nanjing, China). IL-33-deficient (*il-33⁻/⁻*) C57BL/6 J mice with replacement of exon 2 with a tandemly arrayed promoter-less GFP gene and a floxed neomycin resistance gene

**Fig. 4 | Loss of IL-33-mediated thymic involution rescues effective T-cell immunity to control severe infection. a, b** CFSE MFI of naive CD4$^+$ T cells from schistosome-infected mice (WT or WT + Sj, $n = 3$ mice; $il33^{-/-}$, $n = 4$ mice; $il33^{-/-}$+Sj, $n = 5$ mice), $P < 0.0001$. **c, d** CFSE MFI of naive CD4$^+$ T cells from schistosome-infected mice treated with anti-IL-33 ($n = 4$ mice), $P < 0.0001$. **e, f** CFSE MFI of naive CD4$^+$ T cells from mice treated with IL-33 ($n = 3$ mice). $P = 0.0008$, Unpaired two-tailed Student's $t$-test. **g–o** Mice were transplanted with $il1rl1^{+/+}$ or $il1rl1^{-/-}$ thymus. **g** CD5 MFI on naive CD4$^+$ T ($il1rl1^{-/-}$thymus+PBS, $n = 7$ mice; other groups, $n = 8$ mice; pool of two independent experiments). IL-33 versus PBS, $P < 0.0001$ (Spleen or Blood); $il1rl1^{-/-}$thymus+IL-33 versus IL-33, $P = 0.0003$ (Spleen), $P = 0.0205$ (Blood). **h, i** CFSE MFI of naive CD4$^+$ T cells ($n = 8$ mice, pool of two independent experiments), $P < 0.0001$. **j** CD5 MFI on naive CD4$^+$ T ($n = 4$ mice). WT thymus +Infected versus WT thymus+Uninfected, $P < 0.0001$ (Donor thymus, Spleen, or Blood); $il1rl1^{-/-}$thymus+Infected versus WT thymus+Infected, $P < 0.0001$ (Donor thymus), $P = 0.0035$ (Spleen), $P = 0.0899$ (Blood). **k, l** CFSE MFI of naive CD4$^+$ T cells ($il1rl1^{-/-}$ thymus+Infected, $n = 3$ mice; other groups, n $=$ 5 mice), $P < 0.0001$. **m** Representative image of histology of liver, Scale bar, 100 μm. **n** The areas of granulomas around a single egg ($n = 4$ mice). $il1rl1^{+/+}$ thymus+Infected versus Infected or $il1rl1^{-/-}$ thymus+Infected, $P = 0.9535$ or $P < 0.0001$. **o** Survival curves of CLP-operated mice ($n = 8$ mice). WT + $il1rl1^{+/+}$ thymus versus WT or WT + $il1rl1^{-/-}$ thymus, p = 0.3181 or $P = 0.0205$; WT + $il1rl1^{-/-}$ thymus versus WT, $P = 0.0038$; Log-rank (Mantel-Cox) test. All data are shown as the mean ± s.d. *$P < 0.05$, **$P < 0.01$, ***$P < 0.001$, NS, not significant. **b, d, g, i, j, l, n** One-way ANOVA with Tukey's multiple comparisons. Source data are provided as a Source Data file.

and ST2-deficient ($il1rl1^{-/-}$) C57BL/6 J mice with deletion of exon 3 were obtained from Dr. Hong Zhou (Anhui Medical University, Hefei, China). *Pou2f3$^{-/-}$* mice with deletion of exon 3 were obtained from Dr. Minjun Ji (Nanjing Medical University, Nanjing, China). We used six to eight-week-old male mice for all of the experiments. All mice were housed under 20–22 °C with a 12 h light/dark cycle and specific pathogen-free with humidity between 40% and 60% in the Animal Core Facility of Nanjing Medical University.

### Schistosome infection
Mice were infected percutaneously with $18 ± 2$ *S. japonicum* cercariae obtained from infected *Oncomelania hupensis* snails purchased from the National Institute of Parasitic Diseases in Shanghai, China. Unless specifically noted otherwise, normal mice or mice after thymus implantation were sacrificed 8 weeks after infection to investigate the morphology, weight, cellularity, and architecture of the thymus, development, and apoptosis of thymocytes, expression levels of CD5 on thymocytes and peripheral naive T cells, and proliferation of peripheral naive CD4$^+$ T cells from the spleen.

### Schistosome-infected mice treated with praziquantel (PZQ) or anti-IL-33 neutralizing antibody
PZQ (Sigma-Aldrich, St Louis, MO) was administered by gastric gavage with a single dose of 250 mg/kg sodium carboxymethyl cellulose (vehicle) at week 8 after infection. Mice were sacrificed 7 weeks after antischistosomal treatment with PZQ to investigate the morphology, weight, and cellularity of the thymus and the development of thymocytes. For blocking IL-33 signaling, mice at week 3 after schistosome infection were injected intraperitoneally every other day with an anti-IL-33 neutralizing antibody (3.6 ug/mouse, R&D Systems, AF3626) until week 8 after infection.

### Cecal ligation and puncture model
Mice were anesthetized by intraperitoneal administration of 30 μl of ketamine (75 mg/kg) and xylazine (15 mg/kg) diluted in phosphate-buffered saline (PBS). A double puncture was made through the cecum, which was ligated with a 6.0 silk suture at its base below the ileocecal valve with a 19-gauge needle to induce severe CLP sepsis (distance of <1 cm from the distal end of the cecum to the ligation point). On day 9 after CLP induction, mice were sacrificed to investigate the morphology, weight, and cellularity of the thymus and the development of thymocytes.

### Human data
We reviewed the chest computed tomographic (CT) images of all children with sepsis who underwent chest CT examinations ($n = 11$; male/female, 3/8) from January 2020 to November 2020 at the Children's Hospital of Nanjing Medical University. Children with funnel chest but without infectious diseases or tumors who underwent chest CT examinations served as controls ($n = 9$; male/female, 6/3). The age of all participants in this study is from 2 to 8 years old. The thymus index, a marker of thymus size, was measured by a Philips Brilliance 128iCT scanner with IntelliSpace IX Workstation (Philips Healthcare, Andover, MA). The electronic medical records of children were reviewed for demographic and clinical data. Comparisons of age between groups were performed with the Mann–Whitney $U$ test. The Pearson chi-square test was used to test for differences in gender between groups.

### Thymus transplantation
Wild-type or $il1rl1^{-/-}$ embryos were used as thymus tissue for transplantation. Thymus lobes were cultured for 5 days in the presence of 1.35 mM deoxyguanosine (dGUO; Sigma-Aldrich) before transplantation under the kidney capsule of adult nude or wild-type recipients. Eight weeks after thymus implantation, the mice were intraperitoneally injected with recombinant murine IL-33 (1.0 μg/mouse) in PBS (Biolegend, 580508) for six consecutive days and sacrificed 24 h after the last injection to investigate the morphology, weight, and cellularity of the thymus, expression levels of CD5 on thymocytes and peripheral naive T cells, and proliferation of peripheral naive CD4$^+$ T cells from the spleen.

### Fetal thymus organ culture (FTOC)
Thymic lobes were isolated from 15-day embryos (day E15) and cultured for 4 days on nucleopore filters placed in R10 medium containing RPMI 1640 supplemented with 10% fetal bovine serum (FBS), 2 mM ʟ-glutamine, 100 U/ml penicillin, 100 mg/ml streptomycin, and 50 mM 2-mercaptoethanol in the presence of recombinant IL-33 (1 μg/ml; BioLegend, San Diego, CA) or NIK inhibitor (NIKi, 10 μg/ml; MedChemExpress, Shanghai, China). For 2-deoxyguanosine (dGUO) /FTOC, thymic lobes were cultured for 5 days in the presence of 1.35 mM dGUO and then cultured in FTOC in the presence of IL-33 or NIKi for an additional 4 days.

### Flow cytometry
At the indicated time points, single-cell suspensions were prepared from the spleen, thymus, or peripheral blood of individual animals. For analysis of TEC, thymus from schistosome-infected mice or IL-33-treated mice were cut into small pieces, followed by incubation at 37 °C for 15 min and shaking with 50 RPM at 37 °C in 0.5 ml RPMI containing 20 μl of Liberase TH (28 U/ml, Sigma-Aldrich, St. Louis, MO) and DNase1 (5 mg/ml, StemCell Technologies, Vancouver, BC, Canada). Single-cell suspensions from the spleen, thymus, or peripheral blood were blocked with anti-mouse CD16/32 (Invitrogen, 14-0161-86, 93, 1:100) before being stained with the following antibodies: CD3-FITC (Invitrogen, 11-0031-82, 145-2C11, 1:400), CD3-V450 (bdbiosciences, 560804, 500A2, 1:500), CD4-PE-Cy7 (Invitrogen, 25-0042-82,

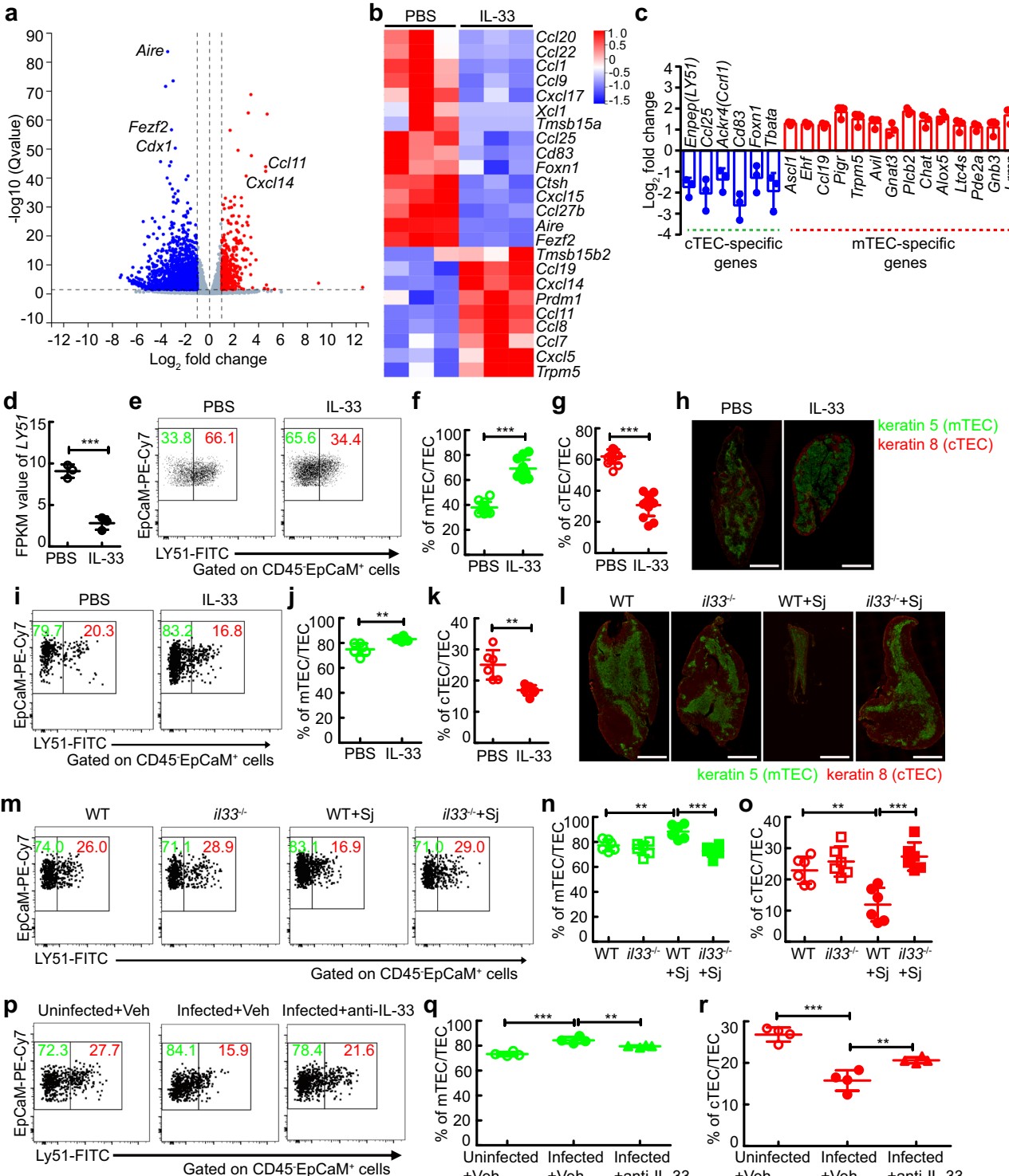

**Fig. 5 | IL-33 perturbs the function and compartment of thymic epithelial cells. a–d** TECs from thymus cultured in FTOC treated with IL-33 (*n* = 3 biologically independent samples of fetal thymic lobes) to perform RNA-seq analysis. **a** Volcano plot shows genes differentially expressed in TECs. **b** Heat map shows TEC function-associated genes. **c** cTEC or mTEC-specific genes. **d** FPKM value of *Enpep* gene (encoding LY51) of TECs (*n* = 3 biologically independent samples of fetal thymic lobes), *P* = 0.0006. **e–g** Representative and quantified flow cytometry of mTECs and cTECs from thymus cultured in FTOC treated with IL-33 (*n* = 13 biologically independent samples of fetal thymic lobes, pool of three independent experiments), *P* < 0.0001 (mTEC or cTEC). **h** Histology of the thymus from mice treated with IL-33 or PBS; Green, Keratin 5; Red, Keratin 8; Scale bar, 1000 μm. **i–k** Representative and quantified flow cytometry of cTECs or mTECs in the thymus from mice treated with IL-33 (*n* = 6 mice, pool of two independent experiments), *P* = 0.0025 (mTEC), *P* = 0.0025 (cTEC). **l** Histology of the thymus from WT or *il33*⁻/⁻ mice 8 weeks after infection; Green, Keratin 5; red, Keratin 8; Scale bar, 1000 μm. **m–o** Representative and quantified flow cytometry of cTECs or mTECs in the thymus from WT or *il33*⁻/⁻ mice 8 weeks after infection (*n* = 6 mice, pool of two independent experiments). WT + Sj versus WT, *P* = 0.0057 (mTEC), *P* = 0.0038 (cTEC); WT + Sj versus *il33*⁻/⁻+Sj, *P* = 0.0002 (mTEC), *P* = 0.0001 (cTEC). **p–r** Representative and quantified flow cytometry of cTECs or mTECs in the thymus from schistosome-infected mice treated with anti-IL-33 (*n* = 4 mice). Infected +Veh versus Uninfected+Veh, *P* < 0.0001 (mTEC), *P* < 0.0001 (cTEC); Infected+Veh versus Infected+anti-IL-33, *P* = 0.0099 (mTEC), *P* = 0.0099 (cTEC). All data are shown as the mean ± s.d. *\*P* < 0.05, *\*\*P* < 0.01, *\*\*\*P* < 0.001, NS, not significant. **d, f, g, j, k** Unpaired two-tailed Student's *t*-test; **n, o, q, r** One-way ANOVA with Tukey's multiple comparisons. Source data are provided as a Source Data file.

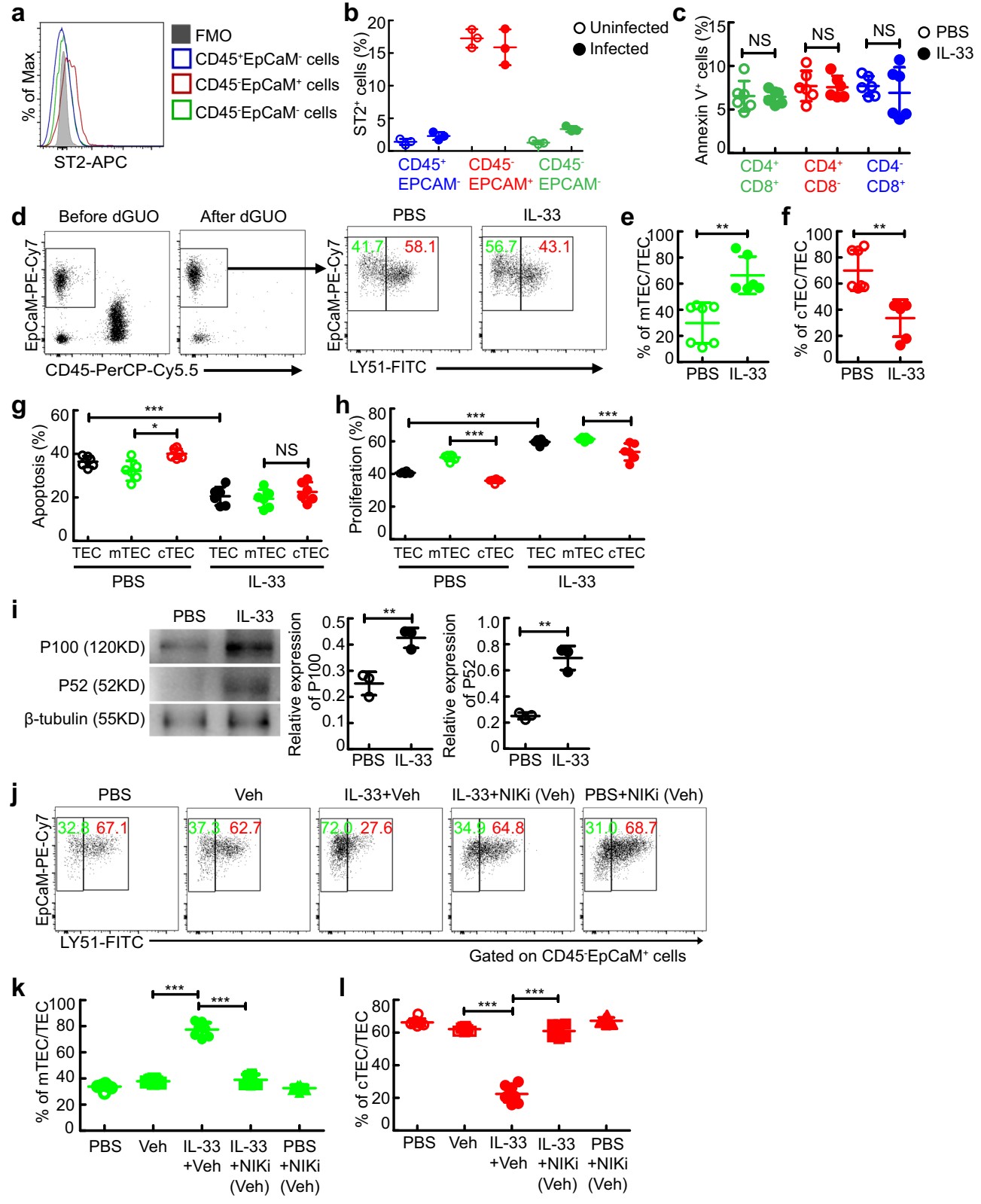

RM4-5, 1:500), CD4-FITC (Invitrogen, 11-0042-85, RM4-5, 1:400), CD25-PE-Cy7 (Invitrogen, 25-0251-82, PC61.5, 1:400), Foxp3-PE (Invitrogen, 12-5773-82, FJK-16s, 1:40), CD8-APC (Invitrogen, 17-0081-82, 53-6.7, 1:1000), CD8-FITC (Invitrogen, 11-0081-82, 53-6.7, 1:200), CD44-PE (Invitrogen, 12-0441-82, IM7, 1:500), CD62L-V450 (BDbiosciences, 562910, MEL-14, 1:400), CD5-PerCP-Cy5.5 (Invitrogen, 45-0051-80, 53-7.3, 1:1000), Annexin-V-APC (Invitrogen, BMS306APC-100, 1:100), Ki-67-Alexa Fluor 647 (bdbiosciences, 558615, B56, 1:100), Ly51-FITC (Invitrogen, 11-5891-82, 6C3, 1:500), CD45-PerCP-Cy5.5 (Invitrogen, 45-0451-82, 30-F11, 1:1000), EpCAM-PE-Cy7 (Invitrogen, 25-5791-80, G8.8,

**Fig. 6 | IL-33 leads to aberrant accumulation of mTECs independently of thymocytes. a, b** Representative and quantified flow cytometry of ST2$^+$ cells gated on CD45$^-$EpCaM$^-$, CD45$^-$EpCaM$^+$, or CD45$^-$EpCaM$^-$ cells in the thymus ($n = 3$ mice). **c** Quantification of flow cytometry of annexin V$^+$ cells from the purified thymocytes treated with IL-33 for 3 days ($n = 6$ mice, pool of two independent experiments). $P = 0.9057$ (CD4$^+$CD8$^+$), $P = 0.8832$ (CD4$^+$CD8$^-$), $P = 0.5614$ (CD4$^-$CD8$^+$). **d–f** Representative and quantified flow cytometry of cTECs or mTECs from the thymus in dGUO-treated FTOCs treated with IL-33 (PBS or IL-33, $n = 7$ or 6 biologically independent samples of fetal thymic lobes, pool of two independent experiments). $P = 0.0011$ (mTEC), $P = 0.0011$ (cTEC). **g, h** Quantification of flow cytometry of annexin V$^+$ cells or Ki67$^+$ cells, gated on TEC, cTEC, or mTEC from the thymus cultured in FTOC treated with IL-33 ($n = 6$ biologically independent samples of fetal thymic lobes, pool of two independent experiments). IL-33+TEC versus

PBS + TEC, $P < 0.0001$ (Apoptosis), $P < 0.0001$ (Proliferation); PBS + mTEC versus PBS + cTEC, $P = 0.0156$ (Apoptosis), $P < 0.0001$ (Proliferation); IL-33+mTEC versus IL-33+cTEC, $P = 0.7150$ (Apoptosis), $P = 0.0001$ (Proliferation). **i** Representative and quantified western blots of P100/P52 in TECs from the thymus cultured in FTOCs treated with IL-33 ($n = 3$ biologically independent samples of fetal thymic lobes). $P = 0.0067$ (P100), $P = 0.0013$ (P52). **j–l** Representative and quantified flow cytometry of mTECs or cTECs from the thymus in FTOCs treated with IL-33 and/or NIK inhibitor (NIKi; $n = 8$ biologically independent samples of fetal thymic lobes, pool of two independent experiments). IL-33+Veh versus Veh, $P < 0.0001$ (mTEC or cTEC); IL-33+Veh versus IL-33+NIKi (Veh), $P < 0.0001$ (mTEC or cTEC). **c, e, f, i** Unpaired two-tailed Student's $t$-test; **g, h, k, l** One-way ANOVA with Tukey's multiple comparisons. All data are shown as the mean ± s.d; *$P < 0.05$, **$P < 0.01$, ***$P < 0.001$, NS, not significant. Source data are provided as a Source Data file.

1:1000), ST2-APC (Invitrogen, 17-9335-82, RMST2-2, 1:150), MHC-II-Brilliant Violet 421 (BioLegend, 107632, M5/114.15.2, 1:1000), ITGB4-FITC (BioLegend, 123606, 346-11 A, 1:100), L1CAM-Alexa Fluor 647 (R&D Systems, FAB5674R-100UG, 1:40), and Ly6D-PE (Invitrogen, 12-5974-80, 49-H4, 1:1000). Cell acquisition was performed using a FACSVerse cytometer (BD Biosciences) with BD FACSuite software (Version 1.0.6.5230). Data were analyzed with FlowJo (Tree Star, version 10.0.7).

### IL-33 administration
Recombinant murine IL-33 (1.0 μg/mouse) in PBS (Biolegend, 580508) was intraperitoneally injected into WT or *il1rl1$^{-/-}$* mice for six consecutive days and analyzed 24 h after the last injection. Control mice were injected intraperitoneally with PBS.

### Proliferation assay (CFSE)
Naive splenic CD4$^+$ T lymphocytes were isolated as CD62L$^+$CD4$^+$ T cells by sorting total splenocytes with a Naive CD4$^+$ T Cell Isolation Kit (Miltenyi Biotec GmbH, Bergisch Gladbach, Germany). Naive CD4$^+$ T cells were resuspended in PBS at $2.5 \times 10^6$/mL for staining. CFSE in the form of a 5 mM stock solution in dimethyl sulfoxide was added at a final concentration of 2.5 μM for 25 min at 37 °C. Naive CD4$^+$ T cells were then washed twice in RPMI. Naive CD4$^+$ T cells ($2 \times 10^5$/well) were cultured in 96-well plates coated with 0.3 μg of anti-CD28 (BioLegend, 102116, 37.51) and 0.1 μg of anti-CD3 antibodies (BioLegend, 100238, 17A2). After 72 h or 96 h of incubation, CD4$^+$ T cells were harvested for analysis.

### Histology and immunofluorescence microscopy
Tissues were fixed in paraformaldehyde, embedded in paraffin, sectioned, and stained with Keratin 5 (Abcam, ab52635, EP1601Y, 1:100), Keratin 8 (DSHB, TROMA-I, NA, 1:10), Alexa Fluor 555 conjugate anti-rat IgG (Cell Signaling Technology, 4417 S, 1:500), or Alexa Fluor 488 conjugate anti-rabbit IgG (Cell Signaling Technology, 4412 S, 1:500) antibodies. The slides were examined under a fluorescence microscope (panoramic MID, Nikon Corporation, Tokyo, Japan), and the images were captured with Pannoramic Scanner software (Version 3.0.2). Paraffin-embedded liver sections were dewaxed and stained with hematoxylin and eosin (H&E) for granuloma analysis as previously described[57].

### Western blots
Thymus or liver from mice, naive splenic CD4$^+$ T cells isolated by a Naive CD4$^+$ T Cell Isolation Kit (Miltenyi Biotec GmbH), or TECs sorted from fetal thymus cultured in FTOC using CD45 MicroBeads (Order No. 130-052-301, Miltenyi Biotec) and EpCAM MicroBeads (Order No. 130-105-958, Miltenyi Biotec) via automated magnetic separation (Miltenyi Biotec) were resolved by 10% (m/v) SDS-PAGE under reducing conditions and transferred to nitrocellulose membranes. After blocking with 2% BSA, the membranes were washed

and incubated with anti-IL-33 (R&D Systems, AF3626, 0.4 μg/mL), P100/P52 (Cell Signaling Technology, 4882 S, 1:1000), Pou2f3 (Novus Biologicals, NBP2-94551, 1:1000), FoxO1 (Cell Signaling Technology, 2880 S, C29H4, 1:1000), GAPDH (Abcam, ab181602, EPR16891, 1:10000), β-tubulin (Cell Signaling Technology, 2146 S, 1:1000), or anti-β-actin (Cell Signaling Technology, 4970 S, 13E5, 1:1000) antibodies overnight at 4 °C. The membranes were washed and incubated with HRP conjugated anti-goat IgG (KPL, 5220-0362, 1:1000), HRP conjugated anti-mouse IgG (Cell Signaling Technology, 7076 S, 1:1000), HRP conjugated anti-rabbit IgG (Cell Signaling Technology, 7074 S, 1:1000) secondary antibodies for 1 h at room temperature and exposed to ECL reagents (Bio-Rad, Hercules, CA).

### Real-time PCR
Naive splenic CD4$^+$ T cells were isolated by a Naive CD4$^+$ T Cell Isolation Kit (Miltenyi Biotec GmbH) and then were lysed in TRIzol™ Reagent (Invitrogen). Isolated total RNA using a phenol-chloroform extraction was reverse-transcribed with the SuperScript III First-Strand cDNA Synthesis System (Invitrogen). RT-PCRs were performed on the ABI PRISM 7300 Real-Time PCR System (Applied Biosystem) using FastStart SYBR Green Master Mix (Roche Applied Science). Primers used were as follows: Ccl1 (forward, 5′-TTCCCCTGAAGTTTATCCAGTGTT-3′; reverse, 5′TGAACCCACGTTTTGTTAGTTGAG-3′), Ccl2 (forward, 5′-GTCCCTGTCATGCTTCTGGG-3′; reverse, 5′-GCGTTAACTGCATCTGGCTG-3′), Ccl3 (forward, 5′-CCAGCCAGGTGTCATTTTCC-3′; reverse, 5′-CTCAAGCCCCTGCTCTACAC-3′), Il-6 (forward, 5′-GAGGATACCACTCCCAACAGACC-3′; reverse, 5′-AAGTGCATCATCGTTGTTCATACA-3′), Tnf (forward, 5′-CCCTCACACTCAGATCATCTTCT-3′; reverse, 5′-GCTACGACGTGGGCTACAG-3′), Csf1 (forward, 5′-GGTCCTGCAGCAGTTGATCG-3′; reverse, 5′-CTCGGTGGCGTTAGCATTGG-3′), Gzmb (forward, 5′-GACCCAAAGACCAAACGTGC-3′; reverse, 5′-TCTGTAGTTAGCTGCTTTTCATTGT-3′), and Gzmk (forward, 5′-CCGCCCACTGCTACTCTTG-3′; reverse, 5′-TGCAGCAGTGCGAAGCTTTATC-3′).

### ELISA for sST2
The serum was collected from uninfected or schistosome-infected mice. The sST2 concentration was measured with a Mouse ST2/IL-33R Quantikine ELISA kit (R&D Systems, MST200), according to the manufacturer's instructions.

### RNA-sequencing
Naive CD4$^+$ T cells were sorted from the spleen in normal mice or schistosome-infected mice via magnetic separation using the CD4$^+$CD62L$^+$ T Cell Isolation Kit (Miltenyi Biotec). TECs were sorted from fetal thymus cultured in FTOC treated with recombinant IL-33 (BioLegend) for 4 days via automated magnetic separation using CD45 MicroBeads (Miltenyi Biotec) and EpCAM MicroBeads (Miltenyi Biotec). The cDNA library construction, library purification, and RNA sequencing (RNA-seq) were conducted on a BGISEQ platform at Huada

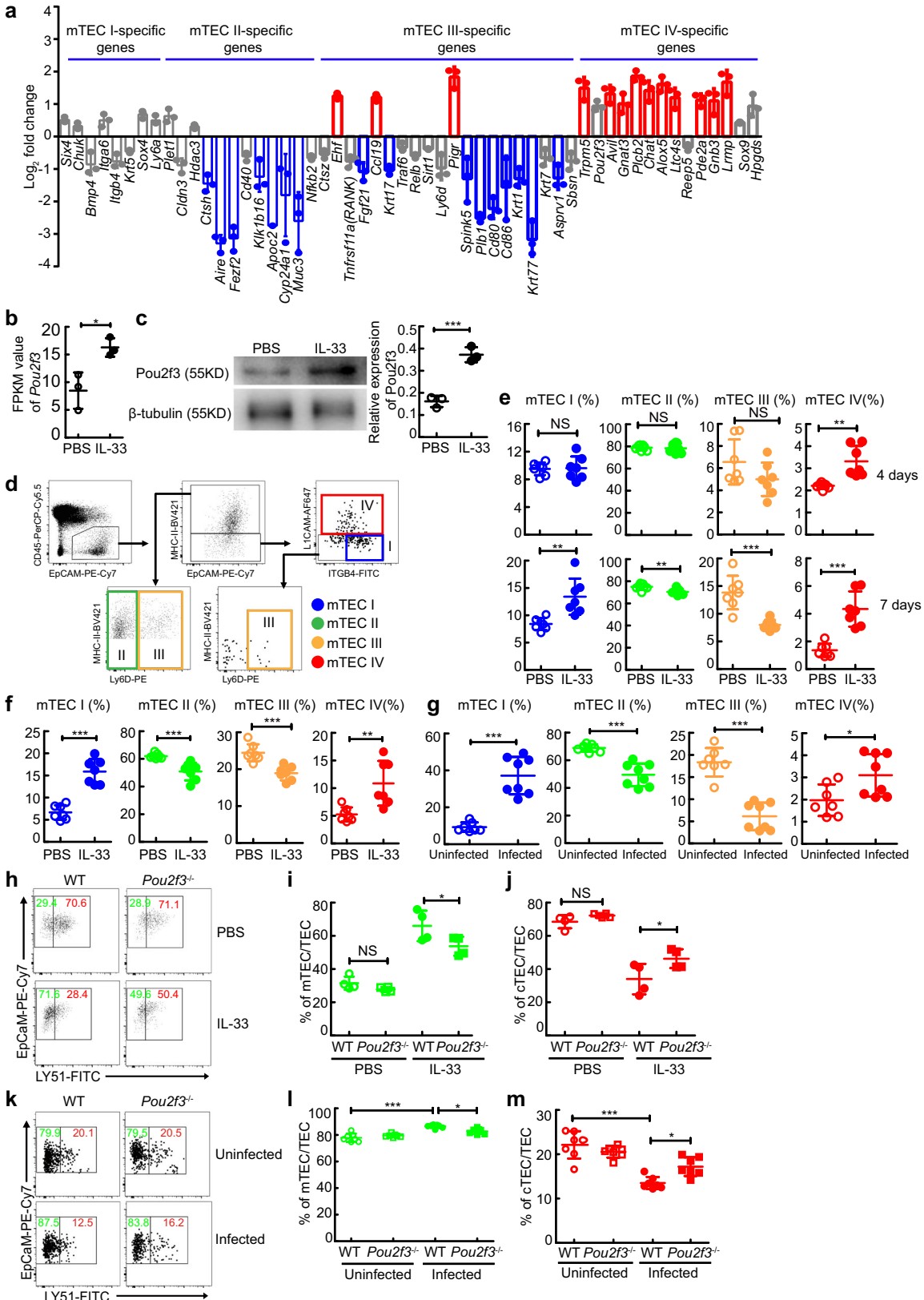

Gene Technology Co., Ltd. (Shenzhen, China) following standard protocols. Standard bioinformatics analysis was performed by BGI. The RNA-seq data of naive CD4+ T cells and TECs were deposited in the NCBI's GEO under accession numbers GSE189200 and GSE189201, respectively.

### Statistics

Statistics and graphing were conducted in GraphPad Prism 5.0 software. Error bars indicate the s.d. Data were analyzed by a Unpaired two-sided Student's *t*-test for two-group comparisons and by One-way ANOVA with Tukey's multiple comparisons or

**Fig. 7 | IL-33 promotes the excessive generation of thymic tuft cells and acute thymic involution in a Pou2f3-dependent manner. a** RNA-seq analysis of mTEC subset-specific genes in TECs in FTOCs treated with IL-33. **b** FPKM values of *Pou2f3* genes (*n* = 3 biologically independent samples of fetal thymic lobes), *P* = 0.0212. **c** Representative and quantified western blots of Pou2f3 in TECs in FTOCs treated with IL-33 for 4 days (*n* = 3 biologically independent samples of fetal thymic lobes), *P* = 0.001. **d** Gating strategy of mTEC subsets. **e** mTEC subset percentages in FTOCs treated with IL-33 for 4 days or 7 days (*n* = 7 biologically independent samples of fetal thymic lobes, pool of two independent experiments). 4 days or 7 days, *P* = 0.886 or 0.0027 (mTEC I), *P* = 0.7789 or 0.0094 (mTEC II), *P* = 0.1266 or 0.0004 (mTEC III), *P* = 0.0013 or <0.0001 (mTEC IV). **f, g** mTEC subset percentages in IL-33-treated or schistosome-infected mice (Infected, *n* = 8 mice, other groups, *n* = 7

mice, pool of two independent experiments). IL-33 versus PBS or Infected versus Uninfected, *P* < 0.0001 or <0.0001 (mTEC I), *P* = 0.0008 or *P* < 0.0001 (mTEC II), *P* = 0.0004 or <0.0001 (mTEC III), *P* = 0.0043 or 0.0238 (mTEC IV). **h–j** The percentages of mTECs or cTECs in FTOCs treated with IL-33 for 4 days (*n* = 4 biologically independent samples of fetal thymic lobes). PBS, *P* = 0.7988 (mTEC or cTEC); IL-33, *P* = 0.0474 (mTEC or cTEC). **k–m** The percentages of cTECs or mTECs in schistosome-infected mice (*n* = 7 mice, pool of two independent experiments). Infected WT versus Uninfected WT or Infected *pou2f3⁻ᐟ⁻*, *P* < 0.0001 (mTEC or cTEC) or *P* = 0.0146 (mTEC or cTEC). **b, c, e, f, g** Unpaired two-tailed Student's *t*-test; (**i, j, l, m**) One-way ANOVA with Tukey's multiple comparisons. All data are shown as the mean ± s.d. \**P* < 0.05, \*\**P* < 0.01, \*\*\**P* < 0.001, NS, not significant. Source data are provided as a Source Data file.

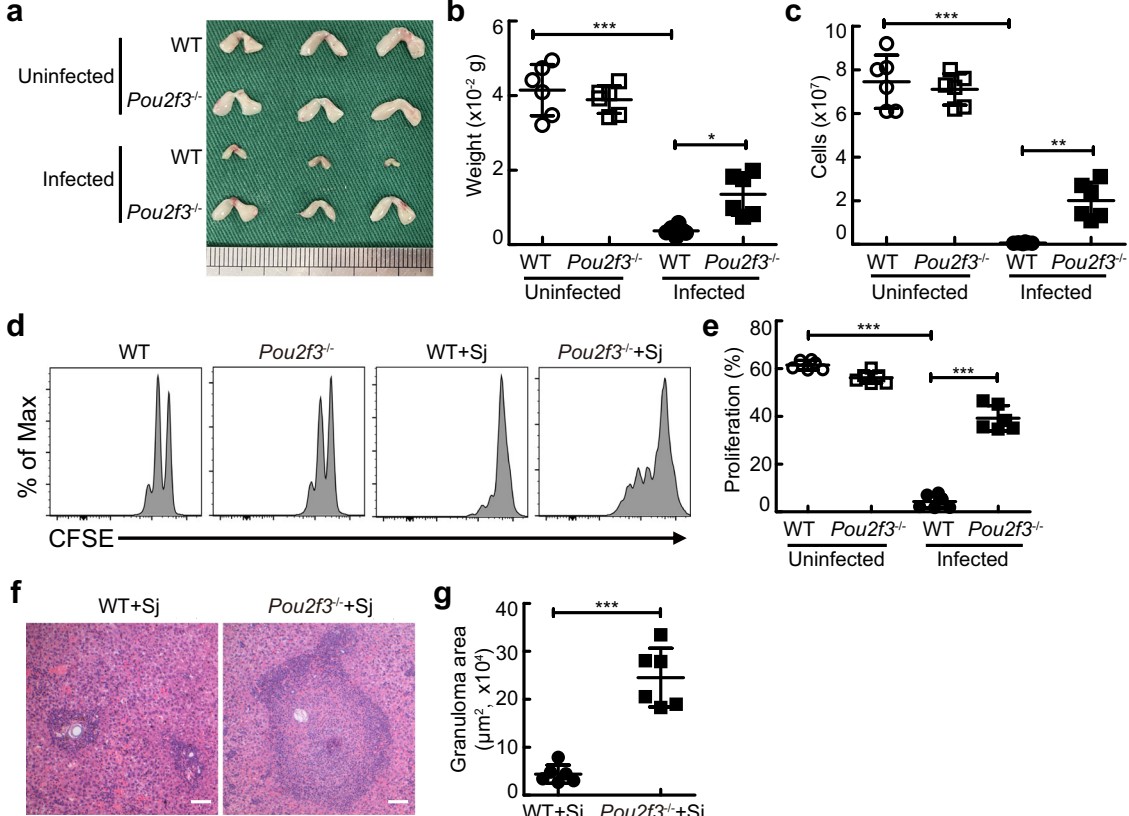

**Fig. 8 | Pou2f3 deficiency abolishes thymic involution-mediated T cell aging during schistosome infection. a–c** Representative morphology and quantified weight and cellularity of thymus from WT or *Pou2f3⁻ᐟ⁻* mice 8 weeks after schistosome infection (*n* = 6 mice, pool of two independent experiments). Infected WT versus Uninfected WT, *P* < 0.0001 (weight), *P* < 0.0001 (cells); Infected WT versus Infected *Pou2f3⁻ᐟ⁻*, *P* = 0.0112 (weight), *P* = 0.0031 (cells); One-way ANOVA with Tukey's multiple comparisons. **d, e** Representative and quantified flow cytometry of CFSE MFI of CFSE-labeled naïve CD4⁺ T cells from uninfected or infected WT or

*Pou2f3⁻ᐟ⁻* mice stimulated with anti-CD3 and anti-CD28 antibodies (*n* = 6 mice, pool of two independent experiments). Infected WT versus Uninfected WT, *P* < 0.0001; Infected WT versus Infected *Pou2f3⁻ᐟ⁻*, *P* < 0.0001; One-way ANOVA with Tukey's multiple comparisons. **f** Representative image of histology of liver from WT or *Pou2f3⁻ᐟ⁻* mice after schistosome infection; Scale bar, 100 μm. **g** The areas of granulomas around a single egg (*n* = 6 mice, pool of two independent experiments), *P* < 0.0001, Unpaired two-tailed Student's *t*-test. All data are shown as the mean ± s.d. \*\**P* < 0.01, \*\*\**P* < 0.001. Source data are provided as a Source Data file.

two-way ANOVA with Tukey's multiple comparisons for comparisons of three or more groups. A *P* value of less than 0.05 was considered significant.

### Reporting summary
Further information on research design is available in the Nature Portfolio Reporting Summary linked to this article.

### Data availability
The RNA-seq data of naïve CD4⁺ T cells and TECs generated in this study have been deposited in the NCBI's GEO under accession codes

GSE189200 and GSE189201, respectively. All other data are included in the Article, Supplementary Information or Source Data file. Source data are provided with this paper.

### Code availability
The full code was available in this article.

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

## Acknowledgements

We thank Minjun Ji from Nanjing Medical University for kindly providing *Pou2f3*$^{-/-}$ mice. This work was supported by the National Key R&D Program of China (2018YFA0507300 to CS), the National Natural Science Foundation of China (grant 81871676 to XJC; grants 82030061 and 81871675 to CS; grant 82102423 to LX), and the Natural Science Foundation of Jiangsu Province (grant BK20190082 to XJC).

## Author contributions

X.J.C., C.S., and S.Z. conceived and designed the experiments. X.J.C., C.W., and L.X. analyzed the data. L.X., C.W., Y.C., Y.W., X.L.S., W.J.C., D.L., H.R.S., W.L., B.B.Y., X.W.W., X.J.Z., Y.X.Y., Z.G.L., R.T., J.F.Z., and Y.L. performed the experiments. The manuscript was written and revised by X.J.C., L.R.L., H.Z., S.Z., and C.S.

## Competing interests

The authors declare no competing interests.
