## [Peer Review File · Nature Communications]

IL-33 induces thymic involution-associated naive T cell aging and impairs host control of severe infectionREVIEWER COMMENTS

Reviewer #1 (Remarks to the Author):

Overview: This manuscript by Xu et al. concludes that infection-induced IL-33 production causes T cell aging by inducing thymic involution to impair host control of severe infections. These investigators also propose that thymic involution is caused by an IL-33 stimulating an imbalance of mTEC to cTEC due to increases in mTEC IV (thymic tuft cells) and subsequent mTEC I cells. The involvement of IL-33 in thymic involution, including that taking place during infection, is well supported. Yet, the conclusion that this results in T cell aging, causing impaired host responses to infection, is not well established. Additional studies and additional controls in current studies are needed to better support these conclusions and bring this quality of this study as a whole to a level of novelty and significance warranting publication in *Nat. Comm.*

Major Concerns:

1. Based on findings presented in Fig. 1 the authors conclude that *Schistosoma japonicum* (Sj) infection causes naive CD4+ T cell aging in mice. These conclusions are based on RNA seq comparing bead isolated CD4+ CD62L+ from B6 mice eight weeks post-injection with Sj to those of uninfected (n=3) mice. These data reveal altered expression of genes associated with T cell aging, such as those ds-DNA repair and damage, senescence, SASP, costimulation, mitochondrial health, and function. It is also asserted that CD5 levels on naive T cells and reduced proliferative capacity further support this age phenotype. The data provided do not convincingly show that the assessed naive T cells are prematurely aged due to conditions resulting from infection. Additional characterization of the T cell compartment (i.e. proportions of naive vs memory; mitochondrial function, reduced repertoire, diminished capacity for polarization, augmented production of pro-inflammatory cytokines; Mittelbrunn and Kroemer. *Nat. Rev. Imm.* 2021. 22: 687-680) between Sj infected versus age-matched non-infected mice is necessary to demonstrate accelerated aging.

2. Based on findings found in Figures 2,3, and 4, as well as Supplemental Figures 2 and 3, the conclusion is put forth that infections (parasitic and bacteria) augmented IL-33 results in immunosuppression by inducing thymic involution that leads to naive T cell aging and impaired host control of schistosomiasis or sepsis. This mechanism is provocative and a role for IL-33 in thymic involution is supported by some solid data, particularly the transplant studies in Figure 3L-M. However, there are numerous weaknesses in these data that need to be resolved.

a. Figure 2A presents a Western blot showing increased IL-33 in the thymus at 8 weeks post Sj injection, but a lack of IL-33 at baseline and the rapid loss of IL-33 following treatment of Sj infection. The Western blot data need to be quantitated across multiple experiments and it should be made clear if it is just the full length (32Kda) or short, more active forms of IL-33 (16-19KDA) that are increased. Given the investigators have an IL-33 deficient mouse, these should also be included as an ideal negative control. The release mechanism(s) of IL-33 to stimulate involution should be speculated on.

b. The concern that there is no convincing evidence provided for augmented T cell aging persists here as well. Again, only limited examples of altered proliferation to polyclonal stimuli are provided. No convincing evidence that IL-33-induced changes contribute to Sj immunity is provided.

c. The switching between several models dilutes and confuses the story. Likewise, the term "severe" infection is often used, but never defined in regards to Sj infection and sepsis. Even the distinct kinetics of disease pathology (death by day 8 versus assessment of thymus and T cell alterations at 8 weeks) makes it difficult to accept that similar immunobiology is underlying these distinct outcomes.

3. The studies present in Figure 5-7 are described to establish that IL-33 acts directly on CD45-EpCAM+ TECs to cause the aberrant accumulation of mTECs that leads to thymic involution. To further support this conclusion there are several items that need addressed.

a. Figure 5. Both in vitro and then in vivo shows that there is an effect of IL-33 on thymic epithelial cells. Yet, the investigators should make better use of the *il1r1*^{-/-} mice to support their findings.

b. The investigators discuss the use of IL-33 targeting agents. There are neutralizing antibodies or

fusion proteins available to deliver in mouse models and define if limiting bioavailable IL-33 during Sj infections can prevent or correct altered mTEC/cTEC balance to reverse thymic involution, and limit naïve T cells aging as predicted.

Minor Concern:

1. In Figure 1 b) and d) uninfected is spelled wrong on the top figure legend.
2. Consistent presentation of representative flow cytometric staining (for example currently lacking for CD5; F-G) and quantification (lacking for proliferation in H) across multiple experiments is necessary throughout to justify conclusions.

Reviewer #2 (Remarks to the Author):

In this manuscript, Xu and colleagues identify a novel role for IL-33 in regulating thymic involution in models of schistosomiasis and sepsis. The authors show that there is severe thymus involution and dysfunction in naïve T cells after infection with schistosomes and show that this is mediated by IL-33 signaling, largely through expansion of Tuft cells and contraction of cTECs. This role of IL-33 signaling in acute thymic involution is very interesting and novel and overall the studies were well designed and executed. The IL-33 experiments in particular are carefully performed, with special credit for the thymus transplantation studies using recombinant IL-33 and *Il1rl1*^{-/-} thymuses as well as the CT examination of pediatric patients with or without sepsis. However, despite these clear strengths, the manuscript also contains some weaknesses; especially around the linking of infection and aging phenotypes of naïve T cells and in the relationship between thymus involution and naïve T cell dysfunction. The following are some major and minor concerns:

Major

1. In Fig. 1 the authors would like to say that infection induces an aging phenotype in naïve T cells but there is no actual comparison to aging mice. Ideally there would also be comparison to naïve T cells that have been derived from old mice. Without that direct comparison, what can be said is that the cells are dysfunctional (or at least,, distinct from baseline) and change expression of some aging-associated genes, but not that they display an aging phenotype. Additionally, since the cells being examined are specifically CD4⁺ T cells, the distinction should be made that this is dysfunction in CD4⁺ T cells, not all T cells as the title of the figure and results sections imply. As it stands the readout for naïve T cell function is proliferation and CD5 expression. Ideally some alternative readouts such as polarization, cytokine production etc. could also be used to strengthen the "infection" phenotype. Given the role of CD4 T cells in regulating schistosome infection, it is possible that the reduced phenotype is because of their increased differentiation/polarization etc. Can the authors comment on this potential interpretation? Given how much it is used, perhaps the authors could also explain a little further the rationale for CD5 as a marker of naïve T cell function.
2. The link between thymus involution and naïve T cell dysfunction is at this point somewhat weak. While there is a clear link between thymus involution and naïve T cells, eventually, it is not entirely clear that is what is happening here as there could be direct effects of infection on naïve T cells in the periphery. The authors show that these dysfunctions are present at 8 weeks after infection (and then a subsequent 7 weeks of anti-schistosome treatment). With acute damage models, thymic involution is followed by a regenerative phase that sees thymic recovery to its pre-damage levels. However, schistosome infection is not cleared so this may be more representative of a chronic damage model. If we knew more about the kinetics of thymus involution as well as changes in peripheral naïve T cells over that time, perhaps useful hints would be gleaned that could have provided a more solid link between involution and naïve T cell dysfunction. This would be especially true if T cell expt of recent thymic emigrants could also be tracked (using either TRECs or RAG2-GFP for instance). The alternative (but likely too complicated experiment that I am not suggesting should actually be done) would be an adoptive transfer of naïve T cells into a nude mouse, which could separate the direct from indirect effects of infection on naïve T cells. Where the link between thymus function and the naïve T cell phenotype has been more clearly established is in the transplant experiments so perhaps they could be leaned on to make that point earlier.
3. In addition to the effect on mTECs, there does seem to be a genuine effect on cTEC function as

well, with decreased Ly51 protein expression at least - not just at the transcription level relative to an expansion of mTECs as could be predicted from the bulk TEC sequencing data. IS there differential expression of cTEC functional markers after IL-33 within cTECs specifically (such as Foxn1, Dll4, etc.)? Is ST2 expressed by both cTECs and mTECs? Data shown in Fig. 6A only shows whole TECs. Furthermore, given we know that Tregs typically express ST2, and there are many recirculating and developing Tregs in the thymus, a more nuanced gating to determine true ST2 expression across the thymus is needed. Given the known effect of IL-33 on Tregs, including in tissue regeneration via amphiregulin (Arpaia et al 2015 Cell), could the authors comment on potential effects of Tregs in these models? Furthermore, soluble ST2 is an important regulator of IL-33 signaling and levels should be shown.

4. With the first half of the manuscript wanting to link thymic involution with naïve T cell dysfunction, and the second half linking IL-33 with expansion of Tuft cells (which drives IL-33/schistosome/sepsis-mediated involution) what happens with naïve T cells in the Pou2f3^{-/-} mice? What about granuloma formation in these mice?

5. In general it would be helpful to make clear the experimental procedures, perhaps with a schema in the graph as some of these experiments are a little

6. It is potentially concerning that all studies are presented showing representative data from one experiment, which means that the datapoints show only equal 3-4 points. Is it possible to merge data from multiple experiments?

Reviewer #4 (Remarks to the Author):

Xu and Wei et al suggest a role for the cytokine IL-33 in promoting T-cell aging and thymic involution in mice following infection with the parasite *Schistosoma japonicum* or by experimentally induced sepsis. Although, I cannot speak to the rigor of the immunological aspects of this manuscript, the work with *S. japonicum* appears to be well executed. Overall, the experiments appear to be well reasoned and the results seem novel and exciting from the perspective of a schistosome biologist. Although the manuscript was readable, it would benefit from editing by a native English speaker.

Minor Comments:

It is important to note the thymic atrophy following schistosome infection has been previously reported: (e.g. <https://journals.asm.org/doi/pdf/10.1128/iai.35.3.1063-1069.1982>)

P5Ln5: Change "proven" to "suggested"

Figure 1B, C, D: what is the scale for the heatmap in 1B, C, D?

Pg 7 Ln 8: response to respond

Pg 7 Ln 16: investigate to investigated

P8Ln16 "the" host immune

P9Ln12 determine "IF"

Figure 4A-C: X axis hard to make out lettering.

We appreciate the helpful and constructive comments of editors and reviewers. As suggested by the reviewers, we have performed additional experiments, added these data into a series of new figures (new Figures 1E-1G, 2A-2F, 3F-3I, 4C-4D, 5P-5R, 6B, 8D-8G, S2A-S2C, S3C-S3F, S3M-S3O, S4D-S4F, S5A-S5D, S6B-S6L, and S7A-S7E), and extensively revised our manuscript. The newly acquired results further support the conclusions described in the manuscript.

Our point-by-point responses to the reviewers' comments are described below:

Reviewer #1 (Remarks to the Author):

Overview: This manuscript by Xu et al. concludes that infection-induced IL-33 production causes T cell aging by inducing thymic involution to impair host control of severe infections. These investigators also propose that thymic involution is caused by an IL-33 stimulating an imbalance of mTEC to cTEC due to increases in mTEC IV (thymic tuft cells) and subsequent mTEC I cells. The involvement of IL-33 in thymic involution, including that taking place during infection, is well supported. Yet, the conclusion that this results in T cell aging, causing impaired host responses to infection, is not well established. Additional studies and additional controls in current studies are needed to better support these conclusions and bring this quality of this study as a whole to a level of novelty and significance warranting publication in Nat. Comm.

Major Concerns:

1. Based on findings presented in Fig. 1 the authors conclude that *Schistosoma japonicum* (Sj) infection causes naive CD4⁺ T cell aging in mice. These conclusions are based on RNA seq comparing bead isolated CD4⁺ CD62L⁺ from B6 mice eight weeks post-injection with Sj to those of uninfected (n=3) mice. These data reveal altered expression of genes associated with T cell aging, such as those ds-DNA repair and damage, senescence, SASP, costimulation, mitochondrial health, and function. It is also asserted that CD5 levels on naive T cells and reduced proliferative capacity further support this age phenotype. The data provided do not convincingly show that the assessed naive T cells are prematurely aged due to conditions resulting from infection. Additional characterization of the T cell compartment (i.e. proportions of naive vs memory; mitochondrial function, reduced repertoire, diminished capacity for polarization, augmented production of pro-inflammatory cytokines; Mittelbrunn and Kroemer. Nat. Rev. Imm. 2021. 22: 687-680) between Sj infected versus age-matched non-infected mice is necessary to demonstrate accelerated aging.

We agree with reviewer #1 and reviewer #2 that the data in figure 1 in the current manuscript were over-interpreted. As suggested by reviewer #2, without actual comparison to naive T cells derived from old mice, naive T cell dysfunction with changed expression of some aging-associated genes, rather than naive T cell aging, might be more appropriate to describe the phenotype of naive T cells in schistosome-infected mice in this manuscript.

As the reviewer pointed out, collectively the characteristic marks of T cell aging are multidimensional and complex (Mittelbrunn and Kroemer. Nat Immunol 2021. 22: 687-698), including ① T cell senescence (impaired proliferation, inhibitory receptors, and senescence markers), ② loss of proteostasis (diminished autophagy and reduced FoxO1 levels), ③

thymic involution (thymocyte reduction, TEC dysfunction, and reduced thymic output), ④ naïve-memory imbalance (increased memory pool, memory inflation, and reduced naïve pool), ⑤ mitochondrial dysfunction (mtDNA reduction, glycolysis, and ROS signaling), ⑥ lack of effector plasticity (extreme differentiation, prevalence of Th1, Th2, Th17, and Tfh, and exhausted and cytotoxic), ⑦ reduced of the TCR repertoire or avidity (reduced TCR diversity, Tscm attrition, and increase in publicity), and ⑧ genetic and epigenetic alterations (DNA damage, telomere attrition, and epigenomic alterations).

We have performed additional experiments to analyze the phenotypes of naïve CD4⁺ T cell aging in schistosome-infected or IL-33-treated mice and added these data into new Figures 1E-1G, 2A-2F, S2A-S2C, and S5A-S5D in our revised manuscript. In conjunction with previous observations, our findings showed that schistosome-infected mice or IL-33-treated mice displayed some characteristic marks of T cell aging, including ① T cell senescence: impaired proliferation (figures 1K-1L and new figures 2E-2F) and senescence markers (figures 1A and 1B), ② loss of proteostasis: reduced FoxO1 levels (new figures 1F and 1G), ③ thymic involution: thymocyte reduction (new figures 2A-1D) and TEC dysfunction (figure 5), ④ naïve-memory imbalance (new figures S5A-S5D), ⑤ mitochondrial dysfunction: glycolysis (figure 1C), ⑥ lack of effector plasticity: senescence-associated secretory phenotype (SASP; figure 1D and new figure 1E), ⑦ reduced of the TCR repertoire or avidity (figures 1H-1J and new figures S2A-S2C), and ⑧ genetic and epigenetic alterations: DNA damage (figure 1B). Indeed, a reduced proportion of naïve-memory T cells was also observed in schistosome-infected mice (see the graph below). However, it may not represent T cell aging because a decrease in naïve T cells may result from infection-triggered activation of naïve T cells, but not T cell aging-related diminished thymic output. Thus, we have decided not to add these data to our revised manuscript. On the other hand, administration of IL-33 alone also resulted in a reduced proportion of naïve-memory T cells in uninfected WT mice, but not in *il1rl1*^{-/-} mice (new Figures S5A-S5D), indicating a possibility that IL-33-mediated thymic involution also contributes to a reduced proportion of naïve-memory T cells during schistosome infection.

(A-D) Representative and quantified flow cytometry of naïve/memory naïve CD4⁺ T cells (A, B) or CD8⁺ T cells (C, D) in the spleen from uninfected or schistosome-infected mice.

However, it is very difficult for us to comprehensively evaluate all the phenotypes of T cell aging that occurred in different circumstances. Considering that this study has focused mainly

on demonstrating the mechanisms of how IL-33 results in thymic involution through inducing thymic tuft cell hyperplasia, which contributes to naïve T cell dysfunction and the impaired control of severe infection, it is plausible that the integrity and quality of our overall work would not be compromised without the comprehensive dissection of phenotypes of naïve T cell aging. Thus, we replaced “naïve T cell aging” with “naïve T cell dysfunction” in our revised manuscript.

2. Based on findings found in Figures 2, 3, and 4, as well as Supplemental Figures 2 and 3, the conclusion is put forth that infections (parasitic and bacteria) augmented IL-33 results in immunosuppression by inducing thymic involution that leads to naïve T cell aging and impaired host control of schistosomiasis or sepsis. This mechanism is provocative and a role for IL-33 in thymic involution is supported by some solid data, particularly the transplant studies in Figure 3L-M. However, there are numerous weaknesses in these data that need to be resolved.
 - a. Figure 2A presents a Western blot showing increased IL-33 in the thymus at 8 weeks post Sj injection, but a lack of IL-33 at baseline and the rapid loss of IL-33 following treatment of Sj infection. The Western blot data need to be quantitated across multiple experiments and it should be made clear if it is just the full length (32Kda) or short, more active forms of IL-33 (16-19KDA) that are increased. Given the investigators have an IL-33 deficient mouse, these should also be included as an ideal negative control. The release mechanism(s) of IL-33 to stimulate involution should be speculated on.

As suggested by the reviewer, we have added the quantitation of western blot data from multiple experiments in our revised manuscript. We have performed additional experiments and found that the cleaved IL-33 (16-19KDa) was significantly increased in the liver (new figure S3A), but not detected in the thymus of schistosome-infected mice (new figure 3A), which may be due to low level of cleaved IL-33 or lack of cleaving proteases of IL-33 in the thymus during schistosome infection. In addition, we have employed IL-33 deficient mouse as a negative control in new figure S3A.

As suggested by the reviewer, we have discussed the mechanisms of IL-33 release in our revised manuscript on page 18 lines 13-22 as follows: IL-33 can be expressed in many types of cells such as epithelial cells, endothelial cells, fibroblasts, macrophages, natural killer T cells, and regulatory T cells [1]. IL-33 lacks a secretion signal and thus does not follow the conventional route of secretion via the endoplasmic reticulum-Golgi apparatus secretory pathway [1]. Besides cellular damage, accumulating evidence support that membrane pores are also involved in the release of IL-33 from cells [1]. For instance, recent reports show that IL-33 can be released from dendritic cells, intestinal epithelial cells, and hepatic stellate cells via membrane pores driven by perforin-2, gasdermin C, and gasdermin D, respectively [2-4]. However, the major cellular source and exact release mechanism of IL-33 during infection-induced thymic involution warrant further investigation.

[1] Dwyer GK, D'Cruz LM, Turnquist HR. Emerging Functions of IL-33 in Homeostasis and Immunity. *Annu Rev Immunol* 40, 15-43 (2022).

- [2] Hung LY, et al. Cellular context of IL-33 expression dictates impact on anti-helminth immunity. *Sci Immunol* 5, (2020).
- [3] Zhao M, et al. Epithelial STAT6 O-GlcNAcylation drives a concerted anti-helminth alarmin response dependent on tuft cell hyperplasia and Gasdermin C. *Immunity* 55, 623-638 e625 (2022).
- [4] Gasdermin D-mediated release of IL-33 from senescent hepatic stellate cells promotes obesity-associated hepatocellular carcinoma. *Sci Immunol* 7, eab17209 (2022).

- b. The concern that there is no convincing evidence provided for augmented T cell aging persists here as well. Again, only limited examples of altered proliferation to polyclonal stimuli are provided. No convincing evidence that IL-33-induced changes contribute to Sj immunity is provided.

We agree with the reviewer that we don't have enough solid data to support the phenotype of naïve T cell aging during schistosome infection. Naïve T cell dysfunction, rather than naïve T cell aging, may be more appropriate to describe the phenotype of naive T cells during schistosome infection in our manuscript.

Schistosome infection-induced immunity is a very complicated process that involves many cells e.g. macrophage, DC, and kinetically orchestrated and multifunctional T cells, including Th1, Th2, Th9, Th17, Tfh, Treg, and CD8⁺ T cells. Although we did not analyze the kinetic changes of T cell-mediated immune responses against schistosomes, thymus transplantation studies using *Il1r1*^{-/-} thymuses sufficed to determine the role of IL-33-induced thymic involution in CD4⁺ T cell response-orchestrated granuloma formation during schistosome infection (Figures 4M and 4N).

- c. The switching between several models dilutes and confuses the story. Likewise, the term "severe" infection is often used, but never defined in regards to Sj infection and sepsis. Even the distinct kinetics of disease pathology (death by day 8 versus assessment of thymus and T cell alterations at 8 weeks) makes it difficult to accept that similar immunobiology is underlying these distinct outcomes.

Severe infection is defined as the infection that results in the systemic inflammatory response, multiple organ tissue damage, and even death in the host, which has been added to our revised manuscript on page 4, line 2. As we know, the prepatent period of schistosome infection is around 5 weeks post-infection. Indeed, schistosome infection causes systemic inflammation, multiple organ damage, and death in mice starting at week 5-6 after infection (Pearce EJ, MacDonald AS. *Nat Rev Immunol* 2002, 2, 499-511).

We have performed additional experiments to evaluate the kinetics of thymic involution as well as changes in peripheral naïve T cells during schistosome infection and added these data into new Figures 2A-2F and S2A-S2C. Results showed that the size and weight of the thymus (new Figures 2A and 2B) and the number of thymocytes (new Figure 2C) were remarkably reduced in mice starting at week 5-6 after schistosome infection, around the time when schistosome eggs provoke a vigorous granulomatous response (Pearce EJ, MacDonald AS. *Nat Rev Immunol* 2002, 2, 499-511). Thymic involution was accompanied by an aberrant

development of T cells (new Figure 2D), decreased expression of CD5 on T cells (new Figures S2A-S2C), and impaired proliferation of peripheral naïve T cells (new Figures 2E and 2F). Thus, excessive inflammation may contribute to thymic involution and impaired T cell proliferation during sepsis or 5-6 weeks after schistosome infection.

3. The studies present in Figure 5-7 are described to establish that IL-33 acts directly on CD45-EpCAM⁺ TECs to cause the aberrant accumulation of mTECs that leads to thymic involution. To further support this conclusion there are several items that need addressed.
 - a. Figure 5. Both in vitro and then in vivo shows that there is an effect of IL-33 on thymic epithelial cells. Yet, the investigators should make better use of the *Il1rl1*^{-/-} mice to support their findings.

As suggested by the reviewer, we have employed the *Il1rl1*^{-/-} mice to further support the roles of IL-33 in thymic involution and the TEC compartment. Results showed that ST2 deficiency abolished IL-33-mediated thymic involution and abnormal TEC compartment in mice injected with IL-33 (new Figures S3M-S3O and S6E-S6H) or infected with schistosome (new Figures S3C-S3E and S6I-S6L). Furthermore, ST2 deficiency also abrogated IL-33-induced abnormal TEC compartment in thymus cultured in FTOC (new Figures S6B-S6D). In addition, ST2 deficiency also rescued IL-33-induced naïve-memory T cell imbalance (new Figures S5A-S5D).

- b. The investigators discuss the use of IL-33 targeting agents. There are neutralizing antibodies or fusion proteins available to deliver in mouse models and define if limiting bioavailable IL-33 during S_j infections can prevent or correct altered mTEC/cTEC balance to reverse thymic involution, and limit naïve T cells aging as predicted.

As suggested by the reviewer, we have employed an anti-IL-33 neutralizing antibody to treat schistosome-infected mice and found that the treatment of the anti-IL-33 neutralizing antibody reversed thymic involution (new Figures 3F-3H), improved T cell development (new Figure 3I), and rescued naïve T cell proliferation (new Figures 4C and 4D) in mice during schistosome infection.

Minor Concern:

1. In Figure 1 b) and d) uninfected is spelled wrong on the top figure legend.

We have corrected these spelling mistakes and double-checked the manuscript.

2. Consistent presentation of representative flow cytometric staining (for example currently lacking for CD5; F-G) and quantification (lacking for proliferation in H) across multiple experiments is necessary throughout to justify conclusions.

As suggested by the reviewer, we have added the representative flow cytometric staining of

CD5 into the new figures 1I, S1A, S2A, S2F, S4A, S4D, S4G, S5E, S5H, and S5J, and the quantification of proliferation from multiple experiments into the new figures 1L, 2F, 2L, 4B, 4D, 4F, 4I, 4L, and 8E in our revised manuscript.

Reviewer #2 (Remarks to the Author):

In this manuscript, Xu and colleagues identify a novel role for IL-33 in regulating thymic involution in models of schistosomiasis and sepsis. The authors show that there is severe thymus involution and dysfunction in naïve T cells after infection with schistosomes and show that this is mediated by IL-33 signaling, largely through expansion of Tuft cells and contraction of cTECs. This role of IL-33 signaling in acute thymic involution is very interesting and novel and overall the studies were well designed and executed. The IL-33 experiments in particular are carefully performed, with special credit for the thymus transplantation studies using recombinant IL-33 and *Il1r1^{-/-}* thymuses as well as the CT examination of pediatric patients with or without sepsis. However, despite these clear strengths, the manuscript also contains some weaknesses; especially around the linking of infection and aging phenotypes of naïve T cells and in the relationship between thymus involution and naïve T cell dysfunction. The following are some major and minor concerns:

Major

1. In Fig. 1 the authors would like to say that infection induces an aging phenotype in naïve T cells but there is no actual comparison to aging mice. Ideally there would also be comparison to naïve T cells that have been derived from old mice. Without that direct comparison, what can be said is that the cells are dysfunctional (or at least,, distinct from baseline) and change expression of some aging-associated genes, but not that they display an aging phenotype. Additionally, since the cells being examined are specifically CD4⁺ T cells, the distinction should be made that this is dysfunction in CD4⁺ T cells, not all T cells as the title of the figure and results sections imply. As it stands the readout for naïve T cell function is proliferation and CD5 expression. Ideally some alternative readouts such as polarization, cytokine production etc. could also be used to strengthen the “infection” phenotype. Given the role of CD4 T cells in regulating schistosome infection, it is possible that the reduced phenotype is because of their increased differentiation/polarization etc. Can the authors comment on this potential interpretation? Given how much it is used, perhaps the authors could also explain a little further the rationale for CD5 as a marker of naïve T cell function.

We agree with the reviewer that the data in figure 1 in the current manuscript were over-interpreted. As suggested by the reviewer, without actual comparison to naïve T cells that have been derived from old mice, naïve T cell dysfunction with changed expression of some aging-associated genes, rather than naïve T cell aging, may be more appropriate to describe the phenotype of naïve T cells in schistosome-infected mice in this manuscript.

We have replaced “naïve T cell aging” with “naïve CD4⁺ T cell aging-associated dysfunction” in the title of figure 1 and result sections.

In addition, we also agree with the reviewer that the reduced phenotype might be because of their increased differentiation/polarization, etc. For instance, although a reduced proportion of naïve-memory T cells was observed in schistosome-infected mice (see the graph below), it may not represent T cell aging because a decrease in naïve T cells may result from infection-triggered activation of naïve T cells, but not T cell aging-related diminished thymic output. Thus, we decided not to add these data to our manuscript. However, we injected IL-33 into uninfected mice and found that administration of IL-33 also caused thymic involution and naïve-memory naïve T cell imbalance (new Figures S5A-S5D), indicating a possibility that IL-33-mediated thymic involution also contributes to a reduced proportion of naïve-memory T cells during schistosome infection.

(A-D) Representative and quantified flow cytometry of naïve/memory naïve CD4⁺ T cells (A, B) or CD8⁺ T cells (C, D) in the spleen from uninfected or schistosome-infected mice.

Since CD5 expression was reported to be developmentally regulated by T cell receptor (TCR) signals and TCR avidity [1], CD5 expression levels have been widely employed to reflect the strength and/or duration of TCR signaling in naïve CD4⁺ T cells and CD8⁺ T cells in the thymus and periphery [2-5]. As suggested by the reviewer, we have added the rationale for CD5 as a marker of naïve T cell function to our revised manuscript on page 7 lines 7-9.

- [1] Azzam HS, Grinberg A, Lui K, Shen H, Shores EW, Love PE. CD5 expression is developmentally regulated by T cell receptor (TCR) signals and TCR avidity. *J Exp Med* 188, 2301-2311 (1998).
- [2] Shinzawa M, et al. Reversal of the T cell immune system reveals the molecular basis for T cell lineage fate determination in the thymus. *Nat Immunol* 23, 731-742 (2022).
- [3] Mandl JN, Monteiro JP, Vriskoop N, Germain RN. T cell-positive selection uses self-ligand binding strength to optimize repertoire recognition of foreign antigens. *Immunity* 38, 263-274 (2013).
- [4] Persaud SP, Parker CR, Lo WL, Weber KS, Allen PM. Intrinsic CD4⁺ T cell sensitivity and response to a pathogen are set and sustained by avidity for thymic and peripheral complexes of self peptide and MHC. *Nat Immunol* 15, 266-274 (2014).
- [5] Kugler DG, et al. Systemic toxoplasma infection triggers a long-term defect in the generation and function of naive T lymphocytes. *J Exp Med* 213, 3041-3056 (2016).

2. The link between thymus involution and naïve T cell dysfunction is at this point somewhat weak. While there is a clear link between thymus involution and naïve T cells, eventually, it is not entirely clear that is what is happening here as there could be direct effects of infection on naïve T cells in the periphery. The authors show that these dysfunctions are present at 8 weeks after infection (and then a subsequent 7 weeks of anti-schistosome treatment). With acute damage models, thymic involution is followed by a regenerative phase that sees thymic recovery to its pre-damage levels. However, schistosome infection is not cleared so this may be more representative of a chronic damage model. If we knew more about the kinetics of thymus involution as well as changes in peripheral naïve T cells over that time, perhaps useful hints would be gleaned that could have provided a more solid link between involution and naïve T cell dysfunction. This would be especially true if T cell export of recent thymic emigrants could also be tracked (using either TRECs or RAG2-GFP for instance). The alternative (but likely too complicated experiment that I am not suggesting should actually be done) would be an adoptive transfer of naïve T cells into a nude mouse, which could separate the direct from indirect effects of infection on naïve T cells. Where the link between thymus function and the naïve T cell phenotype has been more clearly established is in the transplant experiments so perhaps they could be leaned on to make that point earlier.

As suggested by the reviewer, we have performed additional experiments to evaluate the kinetics of thymic involution as well as changes in the functions of peripheral naïve T cells during schistosome infection. We added these data into new Figures 2A-2F and S2A-S2C. Results showed that the size and weight of the thymus (new Figures 2A and 2B) and the number of thymocytes (new Figure 2C) were remarkably reduced in mice starting at week 5-6 after schistosome infection, around the time when schistosome eggs provoke a vigorous granulomatous response (Pearce EJ, MacDonald AS. *Nat Rev Immunol* 2002, 2, 499-511). Thymic involution was accompanied by an aberrant development of T cells (new Figure 2D), decreased expression of CD5 on T cells (new Figures S2A-S2C), and impaired proliferation of peripheral naïve T cells in mice 5-6 weeks after schistosome infection (new Figures 2E and 2F). These results provide a more solid link between involution and naïve T-cell dysfunction.

We agree with the reviewer that transferring TRECs or RAG2-GFP-tracked naïve T cells into nude mice infected with schistosome is a very complicated experiment. Indeed, our results in the thymus transplant experiments have demonstrated that the reduced proliferative capacity of naïve CD4⁺ T cells in schistosome-infected mice was, at least in part, due to thymic involution, as shown by the fact that the transplantation of IL-33 receptor (ST2)-deficient thymus strongly restored T-cell proliferative activity from schistosome-infected mice in figures 4K and 4L. In addition, administration of IL-33 in mice without infection also induced a diminished naïve CD4⁺ T cell proliferation by inducing thymic involution in figures 4H and 4I. Collectively, our results demonstrated that infection had at least an indirect effect on naïve T cells by inducing thymic involution.

3. In addition to the effect on mTECs, there does seem to be a genuine effect on cTEC function as well, with decreased Ly51 protein expression at least - not just at the transcription level relative to an expansion of mTECs as could be predicted from the bulk TEC sequencing data. IS there differential expression of cTEC functional markers after IL-33 within cTECs

specifically (such as Foxn1, Dll4, etc.)? Is ST2 expressed by both cTECs and mTECs? Data shown in Fig. 6A only shows whole TECs. Furthermore, given we know that Tregs typically express ST2, and there are many recirculating and developing Tregs in the thymus, a more nuanced gating to determine true ST2 expression across the thymus is needed. Given the known effect of IL-33 on Tregs, including in tissue regeneration via amphiregulin (Arpaia et al 2015 Cell), could the authors comment on potential effects of Tregs in these models? Furthermore, soluble ST2 is an important regulator of IL-33 signaling and levels should be shown.

As suggested by the reviewer, we have analyzed gene expression of cTEC functional markers and found that the transcriptional levels of several cTEC markers were decreased in IL-33-treated TEC, including *LY51*, *Ccl25*, *Ccr11*, *Cd83*, *Foxn1*, and *Tbata* (new Figure 5C).

We have performed additional experiments to determine ST2 expression on cTEC, mTEC, Treg cells, CD4SP, CD8SP, DP, and DN cells in the thymus and found that TECs, rather than thymocytes, non-TEC stromal cells, or Treg cells, expressed a high level of ST2 (new Figures 6A-6B, and new Figures S7A-S7E). However, the expression level of ST2 was comparable between mTECs and cTECs (new Figures S7B and S7C). We employed deoxyguanosine (dGDUO) to eliminate thymocytes in the thymus and then demonstrated that IL-33 results in aberrant accumulation of mTECs and subsequent thymic involution independently of thymocytes or Treg cells (Figures 3N-3Q and 6D-6F). Although we demonstrated that *pou2f3*-mediated accumulation of mTEC IV contributed to IL-33-induced thymic involution, we can not rule out the possibility that cTECs or non-TEC stromal cells were also involved. The exact contributions of these cells to IL-33-induced thymic involution need to be further investigated.

Although Treg cells were not necessary for IL-33-mediated thymic involution (Figures 3N-3Q and 6D-6F), Treg cells play an important role in IL-33-induced immunoregulation during schistosome infection or sepsis in the peripheral. Our previous published paper shows that IL-33 promotes regulatory T cell accumulation in the liver in mice (Xu L, et al. Eur J Immunol 2018, 48, 1302-1307). Indeed, Treg cells in the liver are reported to play a key role in suppressing egg granuloma formation during schistosome infection (Singh KP, et al. Immunology 2005, 114, 410-417). In addition, IL-33-induced Treg cell expansion is also reported to contribute to long-term immunosuppression during sepsis (Nascimento DC, et al. Nat Commun 2017, 8, 14919). Thus, IL-33 may cause immunosuppression in mice during severe infection, such as sepsis or schistosome infection, by inducing thymic involution and/or Treg cell accumulation.

As suggested by the reviewer, the serum level of soluble ST2 was determined by ELISA in uninfected mice or schistosome-infected mice. The results showed that the serum level of sST2 was significantly increased in mice after schistosome infection (new Figure S3F), indicating a potential regulatory role of sST2 in IL-33-mediated thymic involution.

4. With the first half of the manuscript wanting to link thymic involution with naïve T cell dysfunction, and the second half linking IL-33 with expansion of Tuft cells (which drives IL-33/schistosome/sepsis-mediated involution) what happens with naïve T cells in the *Pou2f3*^{-/-} mice? What about granuloma formation in these mice?

We have performed additional experiments to determine the functions of naïve T cells and the formation of granulomas in *Pou2f3*^{-/-} mice infected with schistosome and found that *Pou2f3* deficiency alleviated thymic involution (Figures 8A-8C), restored naïve T cell proliferation (new Figures 8D and 8E), and promoted T cell-mediated granuloma formation (new Figures 8F and 8G) in mice after schistosome infection. We have added these data into new Figure 8 in our revised manuscript.

5. In general it would be helpful to make clear the experimental procedures, perhaps with a schema in the graph as some of these experiments are a little

We have added the schemas in the graph to make clear the experimental procedures of thymus transplantation in new figure S3P.

6. It is potentially concerning that all studies are presented showing representative data from one experiment, which means that the datapoints show only equal 3-4 points. Is it possible to merge data from multiple experiments?

As suggested by the reviewer, we have merged data from multiple experiments.

Reviewer #4 (Remarks to the Author):

Xu and Wei et al suggest a role for the cytokine IL-33 in promoting T-cell aging and thymic involution in mice following infection with the parasite *Schistosoma japonicum* or by experimentally induced sepsis. Although, I cannot speak to the rigor of the immunological aspects of this manuscript, the work with *S. japonicum* appears to be well executed. Overall, the experiments appear to be well reasoned and the results seem novel and exciting from the perspective of a schistosome biologist. Although the manuscript was readable, it would benefit from editing by a native English speaker.

The manuscript has been edited by native English speakers.

Minor Comments:

1. It is important to note the thymic atrophy following schistosome infection has been previously reported: (e.g. <https://journals.asm.org/doi/pdf/10.1128/iai.35.3.1063-1069.1982>)

We have noted this study and added the citation.

2. P5Ln5: Change “proven” to “suggested”

We have carefully edited the manuscript to make it more concise.

3. Figure 1B, C, D: what is the scale for the heatmap in 1B, C, D?

The genes we showed in figure 1B-1C have been reported as the characteristic marks of T cell aging.

4. Pg 7 Ln 8: response to respond
Pg 7 ln 16: investigate to investigated
P8ln16 “the” host immune
P9Ln12 determine “IF”
Figure 4A-C: X axis hard to make out lettering.

We have carefully edited the manuscript to make it more concise and clear.

REVIEWERS' COMMENTS

Reviewer #1 (Remarks to the Author):

I commend the authors on a very interesting study that now convincingly shows that IL-33 causes thymic involution by directing thymic tuft cell hyperplasia contributes to naïve T cell dysfunction and the impaired control of severe infections. By now using both receptor and ligand knockouts, as well as IL-33 targeting biologics, it is very clear that IL-33, presumable from local sources, but potentially systemically, has surprising immunosuppressive actions during infection. The use of clinical samples from patients suffering from sepsis also provides compelling clinical correlates with the mechanistic rodent findings. These findings have major implications for many areas (trauma, viral and bacterial infections, aging) and I feel this is an impressive and note-worthy study.

Heth R. Turnquist, PhD

Reviewer #2 (Remarks to the Author):

Overall the authors have been extremely responsive and the manuscript is clearly improved. I have just a few minor suggestions to improve readability (several of which were minor comments in my prior review).

- The ST2 staining on TECs is relatively weak, especially in comparison to the known expression of ST2 on Tregs (Fig. S7B vs S7D). There does clearly seem to be some expression (especially compared to the FMO control in Fig. 6A), I am just not convinced it is at a higher level than Tregs, which is what the description to Fig. S7 would like to suggest. This could be because the gates for ST2 expression on TECs is very generous and very likely includes some negative cells. Perhaps MFI could be used as less bias is introduced and also an isotype control is probably a slightly more robust control. At the very least the FMO needs to be shown for the plots in Fig. S7. This is not a major point as it likely does not alter the interpretation but should be made a bit clearer. If this is a point the authors would like to make, perhaps direct comparison with gene expression between Tregs and TECs would be appropriate.
- As per my prior review, I strongly suggest that the colors are changed in the sequencing experiments as green and red together are a very common color blind combination that may make it harder to interpret for a large number of readers (myself included).
- Experimental detail such as timepoints after treatment (for instance, after recombinant IL-33, anti-IL33 etc.) need to be included in figure legends.
- In the text, Fig. S5A is used to call out the FTOC experiment, that should be Fig. S6A.
- Again, as per my previous comments, with the three models used in Fig. 5 to make a similar point perhaps more clearly highlighting this with headings, or some other way to distinguish which data belongs to which model in the actual figure would not only make it easier for the reader, but also more powerful as it clearly demonstrates the same effect in three distinct models. This would also serve to address one of the concerns (2c) of reviewer 1 that may not have been adequately addressed.

We appreciate the helpful and constructive comments of the reviewers and the members of the editorial board. We are pleased that the reviewers considered our findings to be a very interesting, impressive and note-worthy study and that the editors are likely to accept our manuscript for publication. As suggested by the reviewers, we have reanalyzed the data and revised our manuscript to interpret the data more appropriately. In addition, we have carefully edited our manuscript according to both the reviewers' and the editorial requests.

Our point-by-point responses to the reviewers' comments are described below:

REVIEWERS' COMMENTS

Reviewer #1 (Remarks to the Author):

I commend the authors on a very interesting study that now convincingly shows that IL-33 causes thymic involution by directing thymic tuft cell hyperplasia contributes to naïve T cell dysfunction and the impaired control of severe infections. By now using both receptor and ligand knockouts, as well as IL-33 targeting biologics, it is very clear that IL-33, presumable from local sources, but potentially systemically, has surprising immunosuppressive actions during infection. The use of clinical samples from patients suffering from sepsis also provides compelling clinical correlates with the mechanistic rodent findings. These findings have major implications for many areas (trauma, viral and bacterial infections, aging) and I feel this is an impressive and note-worthy study.

Heth R. Turnquist, PhD

We appreciate Dr. Heth R. Turnquist considered our findings to be an interesting, impressive and note-worthy study.

Reviewer #2 (Remarks to the Author):

Overall the authors have been extremely responsive and the manuscript is clearly improved. I have just a few minor suggestions to improve readability (several of which were minor comments in my prior review).

1. The ST2 staining on TECs is relatively weak, especially in comparison to the known expression of ST2 on Tregs (Fig. S7B vs S7D). There does clearly seem to be some expression (especially compared to the FMO control in Fig. 6A), I am just not convinced it is at a higher level than Tregs, which is what the description to Fig. S7 would like to suggest. This could be because the gates for ST2 expression on TECs is very generous and very likely includes some negative cells. Perhaps MFI could be used as less bias is introduced and also an isotype control is probably a slightly more robust control. At the very least the FMO needs to be shown for the plots in Fig. S7. This is not a major point as it likely does not alter the interpretation but should be made a bit clearer. If this is a point the authors would like to make, perhaps direct comparison with gene expression between Tregs and TECs would be

appropriate.

As suggested by the reviewer, we have added the FMO and MFI of ST2 in new figure S7 in our revised manuscript. We found that ST2 MFI was comparable on these subsets of cells in the thymus from schistosome-infected mice (new figure S7f). We have added these data and modified description to Fig. S7 as follows: “Flow cytometry analysis showed that a higher percentage of TECs expressed ST2 compared with thymocytes, non-TEC stromal cells, or Treg cells (Fig. 6a, 6b, and Fig. S7a-S7e), whereas the MFI of ST2 was comparable on these subsets of cells in the thymus during schistosome infection (Fig. S7f).” on page 13 lines 11-14, in our revised manuscript to make it clearer.

2. As per my prior review, I strongly suggest that the colors are changed in the sequencing experiments as green and red together are a very common color blind combination that may make it harder to interpret for a large number of readers (myself included).

We are really sorry about that and as suggested by the reviewer, we changed the colors of the sequencing experiments to blue and red in new figures 1a, 1b, 1c, 5a and 5b in our revised manuscript.

3. Experimental detail such as timepoints after treatment (for instance, after recombinant IL-33, anti-IL33 etc.) need to be included in figure legends.

As suggested by the reviewer, we have added these experimental details in figure legends of figures 3f-3i and 3j-3m.

4. In the text, Fig. S5A is used to call out the FTOC experiment, that should be Fig. S6A.

We would like to thank the reviewer for pointing out this mistake and have corrected it in our revised manuscript.

5. Again, as per my previous comments, with the three models used in Fig. 5 to make a similar point perhaps more clearly highlighting this with headings, or some other way to distinguish which data belongs to which model in the actual figure would not only make it easier for the reader, but also more powerful as it clearly demonstrates the same effect in three distinct models. This would also serve to address one of the concerns (2c) of reviewer 1 that may not have been adequately addressed.

As suggested by the reviewer, we have highlighted this point in the heading as follows: “IL-33 perturbs the compartment of thymic epithelial cells both *in vitro* and in mice with IL-33 administration or severe infection”.